# A feedforward mechanism for human-like contour integration

Fenil R. Doshi [1,2]*, Talia Konkle[1,2], George A. Alvarez[1,2]

1 Department of Psychology, Harvard University, Cambridge, Massachusetts, United States of America,
2 Kempner Institute for the Study of Natural and Artificial Intelligence, Harvard University, Cambridge, Massachusetts, United States of America

* fenil_doshi@fas.harvard.edu

## Abstract

Deep neural network models provide a powerful experimental platform for exploring core mechanisms underlying human visual perception, such as perceptual grouping and contour integration—the process of linking local edge elements to arrive at a unified perceptual representation of a complete contour. Here, we demonstrate that feedforward convolutional neural networks (CNNs) fine-tuned on contour detection show this human-like capacity, but without relying on mechanisms proposed in prior work, such as lateral connections, recurrence, or top-down feedback. We identified two key properties needed for ImageNet pre-trained, feed-forward models to yield human-like contour integration: first, progressively increasing receptive field structure served as a critical architectural motif to support this capacity; and second, biased fine-tuning for contour-detection specifically for gradual curves (~20 degrees) resulted in human-like sensitivity to curvature. We further demonstrate that fine-tuning ImageNet pretrained models uncovers other hidden human-like capacities in feed-forward networks, including uncrowding (reduced interference from distractors as the number of distractors increases), which is considered a signature of human perceptual grouping. Thus, taken together these results provide a computational existence proof that purely feedforward hierarchical computations are capable of implementing gestalt "good continuation" and perceptual organization needed for human-like contour-integration and uncrowding. More broadly, these results raise the possibility that in human vision, later stages of processing play a more prominent role in perceptual-organization than implied by theories focused on recurrence and early lateral connections.

## Author summary

A central challenge in vision science is understanding how the visual system links fragmented local features into coherent object representations. One foundational

---

**Data availability statement:** All data, training code, analysis code, and figure-plotting code is available on Github: https://github.com/feziodoshi/dnn_contour_integration.

**Funding:** This research was funded by the Kempner Graduate Fellowship to FRD, National Science Foundation (NSF CAREER: BCS-1942438) to TK, and National Science Foundation (NSF PAC COMP-COG 1946308) to GAA. The funders had no role in study design, data collection and analysis, decision to publish, or preparation of the manuscript.

**Competing interests:** The authors have declared that no competing interests exist

process supporting this ability is contour integration – the perceptual grouping of aligned edge elements into extended contours. While humans perform this task effortlessly, the underlying computational principles remain unclear. Here, we investigate whether deep neural networks (DNNs) can approximate human-like contour integration and, if so, what computational properties support this ability. We find that while standard object-recognition-trained feedforward CNNs don't exhibit this capacity out-of-the-box, they can be fine-tuned to do so. We identify two key factors that support human-like contour integration in purely feedforward CNNs: a gradual progression of receptive field sizes across layers and a biased sensitivity to gradually curved contours around 20 degrees. We further show that fine-tuning uncovers other human-like capacities in feedforward models, including uncrowding. These findings challenge the view that lateral/recurrent mechanisms are strictly necessary for these abilities, instead demonstrating that hierarchical feedforward processing alone can, in principle, support contour integration. More broadly, our results provide a framework for evaluating when certain grouping computations can emerge in DNNs and what constraints may be necessary to better align artificial vision models with biological vision.

## Introduction

As we shift our gaze from location to location, our perceptual system effortlessly transforms millions of disconnected points of light into coherent representations. These representations support a myriad of downstream tasks, from object recognition to social perception, to visually guided reaching and navigation within dynamic environments. To enable this rich behavioral repertoire, the visual system capitalizes on spatial and temporal regularities to parse and organize the visual field into meaningful holistic units – contours, surfaces, shapes, groups, agents – with predictable behaviors and interaction affordances [1–6].

The process of forming such holistic visual representations was most famously studied by the Gestalt psychologists [7–9] who emphasized the importance of configural relationships between parts, whereby the perceptual system groups together elements that are likely from the same physical object in the environment to represent meaningful, structured "wholes." Gestalt psychologists documented several laws that dictate which elements will be grouped together, such as proximity, similarity, connectedness, continuity, parallelism, closure, and symmetry [10–14]. While these grouping phenomena were initially examined through the lens of subjective perceptual experience, efforts have been made to objectively measure and model them in well-controlled experiments with stimuli that offer parametric control [10,15–18]. However, progress integrating these ideas into a broader theory of vision has been hindered by the lack of image-computable models of high-level visual processing that can manifest these phenomena.

One hundred years after Gestalt Psychology started, deep neural networks (DNNs) have emerged as powerful image computable models that can perform

high-level visual tasks, and thus have the potential to fill this void. DNN models are promising not only because they are highly performant on challenging visual tasks [19,20], but also because they predict human perceptual judgments [21–25] and activity in visual cortex [26–31] better than any previous hand-designed models. Despite these strengths, however, these models appear to lack human-like structure in their internal representations: DNNs favor local features like color and texture over global shape-related features [32–35], lack sensitivity to configural relationships between object parts [36–38], and show limited or mixed success in capturing perceptual grouping phenomena [39–42]. While these shortcomings have been well-documented, it remains unclear the extent to which the lack of structured representations within these models reflects aspects of the visual diet, model architecture, task objective, and/or regularization pressures that shape representations [40,43–45]. For instance, in the case of visual uncrowding, Doerig et al. [42] have suggested that models with purely feed-forward architectures have a fundamental architectural limit and are unable to produce the "uncrowding" signatures found in human perception, where Gestalt grouping mechanisms reduce the effect of crowding on local visual judgements [42,45].

To gain some traction on these questions, we focus our investigation primarily on a specific perceptual grouping operation known as contour integration [46–48]. Contour integration is the process of stitching together local edge representations [48–50], which may appear fragmented in the retinal image due to factors like clutter, occlusion, and contrast differences between the figure and ground ([51]; see **Fig 1A**). Contour integration relies particularly on the *continuity principle* (i.e., gestalt "good continuation") and makes an interesting case study because it is amenable to pychophysical experiments [48], and has been proposed to rely on lateral mechanisms [48,52] as well as feedback and recurrence [53–61]. Moreover, prior work has assumed that these mechanisms are likely required because the combinatorial explosion of possible contour configurations prevent a feed-forward solution [48]—an assumption we can now test with modern DNN models.

Here we examine whether purely feed-forward deep convolutional neural networks can implement contour integration. First, we show that CNNs trained on ImageNet classification do not automatically show strong contour detection abilities,

## A  Perceptual Phenomena

## B  Psychophysical Stimuli to probe continuity


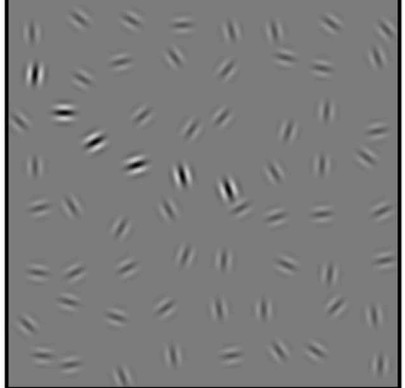

Contour Present Contour Absent

**Fig 1. Visual grouping and contour integration in human perception. (A)** Demonstration of the phenomena where observers can group boundaries from the same object despite occlusions. This panel was generated using OpenAI's DALL.E. **(B)** Psychophysical stimuli to examine the 'continuity' principle underlying contour integration (Field et al., 1993). The left panel displays a contour-present stimulus containing a subset of gabor patches (i.e., contour elements) systematically aligned to suggest an extended, coherent contour amidst randomly oriented background patches. The right panel displays a contour-absent stimulus with identical gabor patches, except the contour patches are also randomly oriented, eliminating the perception of the continuous contour. Contour patches are highlighted for illustrative clarity.

but they can be fine-tuned to do so even without lateral connections, recurrence, or top-down feedback mechanisms. Second, we establish that this capacity is particularly well-supported by the architectural inductive bias of non-linear processing over progressively increasing receptive field sizes. Third, we find that our model quantitively predicts human sensitivity to different global contour curvatures [48] best when tuned to integrate local elements with small orientation differences—a bias towards straight edges or gradual curves. Collectively, these findings provide compelling computational evidence that purely feedforward architectures can support contour integration in a manner that is consistent with human psychophysical data, but that additional inductive biases are needed for models to show a fully emergent alignment with human perception without fine-tuning specifically on contour-detection. More broadly, the finding that a hierarchy of progressively increasing receptive fields enables the extraction of large-scale structure (e.g., an extended contour) from relatively fine-grained properties of the input (e.g., alignment of relatively high frequency elements) suggests that purely feed-forward processing and later stages of the hierarchy can play a more significant role in higher-order grouping operations than prior work has assumed. We propose an integrated view of contour integration in which lateral and recurrent processes refine the inputs to the feed-forward hierarchy, enabling the feed-forward mechanisms to perform the crucial integration operations over distilled inputs.

Finally, we push this approach a step further, examining whether purely feed-forward models can show the uncrowding effect, which is considered a signature of more complex human perceptual grouping operations [42,45]. We find that without fine-tuning feed-forward models do not show uncrowding, consistent with prior work [42,45] but that with fine-tuning they can show human-like uncrowding phenomena—consistent with our contour integration results.

The emergent alignment between the behavior of fine-tuned models and human psychophysical data suggests that both contour integration and more complex perceptual organization mechanisms can be implemented within purely feed-forward deep neural network models, and that fine-tuning pre-trained models is an important methodological approach for uncovering the functional capacities of deep neural network models.

## Results

### Feedforward deep convolutional neural networks (DCNNs) can support fine-grained detection of extended contours

Here we assess how well a classic, feed-forward, deep convolutional neural network architecture – the Alexnet – can support contour detection via contour integration. While these models can successfully categorize objects among a thousand categories, they are infamously "texture-biased" [34], and can even recognize objects successfully even when images are globally scrambled [62]. Additionally, these models lack mechanisms such as within-layer association (self-attention) or lateral recurrence operations, which explicitly enable associative processing of image patches within a layer. Thus, we hypothesized that feed-forward convolutional models, due to these architectural limitations, would perform poorly at the task of detecting an extended contour of local gabor elements hidden amidst a grid of randomly-oriented gabor elements.

To test this idea, we first developed a controlled set of contour stimuli for a contour detection task, drawing directly from prior human psychophysics research [48]. Each display consists of a jittered grid of 256 gabor elements (arranged on a 16 x 16 grid in images of size 512 x 512 pixels). Half the displays contain an embedded contour, where the orientations of 12 local gabor elements are aligned to form an extended path (see **Fig 1B**). We refer to these elements (i.e., the contour elements) as having aligned orientations, because they form paths with varying degrees of global curvature. The remaining background gabor elements in the display are all randomly oriented. These 'contour-present' displays contain a hidden contour. For each contour-present display we also generate matched 'contour-absent' displays which contain identical background gabor elements as their corresponding contour-present images, while the contour elements in these displays are misaligned with random orientations. This careful stimulus construction prevents detecting contours via other cues of proximity or density, requiring integration across the aligned co-circular elements to distinguish contour-present from contour-absent displays. See Methods and S1 Fig for more details on stimuli generation.

To examine the capacity of the Alexnet model's ability to detect these contours, we compared three critical model variants. (1) An untrained model with randomly initialized weights, to test whether the model architecture alone, without training, enables contour integration. (2) A model trained on ImageNet 1000-way image classification, to examine whether the features learned over natural images to support object recognition show improved contour integration. (3) A model that is first trained on ImageNet to do 1000-way classification, and then subsequently fine-tuned end-to-end to support contour integration. This final variant explores the maximum capacity that could be supported by the architecture by directly fine-tuning on the contour detection task. And, for all three models, we probed each layer, to examine whether contour information is more accessible at earlier or later stages of the hierarchy.

To quantify the contour detection capacity of any given model layer, we trained a separate linear classifier over the output activations of each layer to distinguish between the contour-present and contour-absent displays. Stimuli included displays with contour curvatures spanning a broad range of curvature (from gradual to irregular). Classification accuracy was assessed on a held-out set of test images, as our measure of the contour-detection capacity at each layer (see Methods for details on contour-detection linear probing).

The results of all three model variations are shown in Fig 2A. First, the untrained model shows poor contour detection accuracy, which never exceeds 60% for any layer of the model (gray dots). This result indicates that the convolutional inductive bias alone provides minimal support for contour integration. The model trained on ImageNet classification (Fig 2A, orange dots) did show higher contour detection accuracy than untrained models, particularly within deeper layers of the model. This result indicates that the features learned over natural images to support object recognition do in fact improve the contour integration capacity of the Alexnet model, conferring better contour detection at relatively later stages of the hierarchy. However, the contour detection accuracy in the Imagenet trained model peaked at a modest 68%.

Surprisingly, despite the fact that Alexnet is a purely feedforward CNN, the model that was subsequently fine-tuned on the contour detection task led to a substantially greater capacity for contour integration (Fig 2A, blue dots). This was true particularly within the deeper layers, where accuracy is consistently above 90% with a peak of 94.3%. This result reveals that models with purely feed-forward convolutional processing are in fact architecturally-capable of accurately detecting these extended contours.

While our initial plan was to begin adding in other mechanisms that enable convolutional neural networks to do contour integration, we instead explored how it was possible that these fine-tuned feed-forward networks could succeed at the task at all. Note that with the success of these fine-tuned models, we are not proposing a learning story where the visual system learns contour integration through training on the task explicitly (or anything of that sort). Instead, our aim next is to clarify how this purely feed-forward convolutional architecture is succeeding at this task. This is an important research direction for two reasons. First, current theoretical accounts of contour integration generally assume that local associative mechanisms —such as lateral connections or top-down feedback from subsequent processing stages—are *necessary* for contour integration. Therefore, understanding how such a feed-forward model achieves this competence without such mechanisms will reveal new insights for the task of contour detection. Second, it's possible the model's mechanism is also at play in the human visual system and understanding how the model is achieving this could therefore provide new insights into this phenomenon in human vision.

### Fine-tuned networks detect contours via integration of locally aligned elements

How is it that our models trained over natural images, and then fine-tuned on contour detection, can successfully perform the contour detection task by their later processing stages? One potential concern about these results is that these models have found some short-cut or confound present in the stimulus displays, which allows them to perform well on the contour task without actually detecting the contour per se. Do these fine-tuned models (Fig 2A, blue dots) actually integrate information across the entire contour? We first address this potential issue by leveraging tools from the field of feature-visualization, to quantify the extent to which the model isolates the contour elements from the background

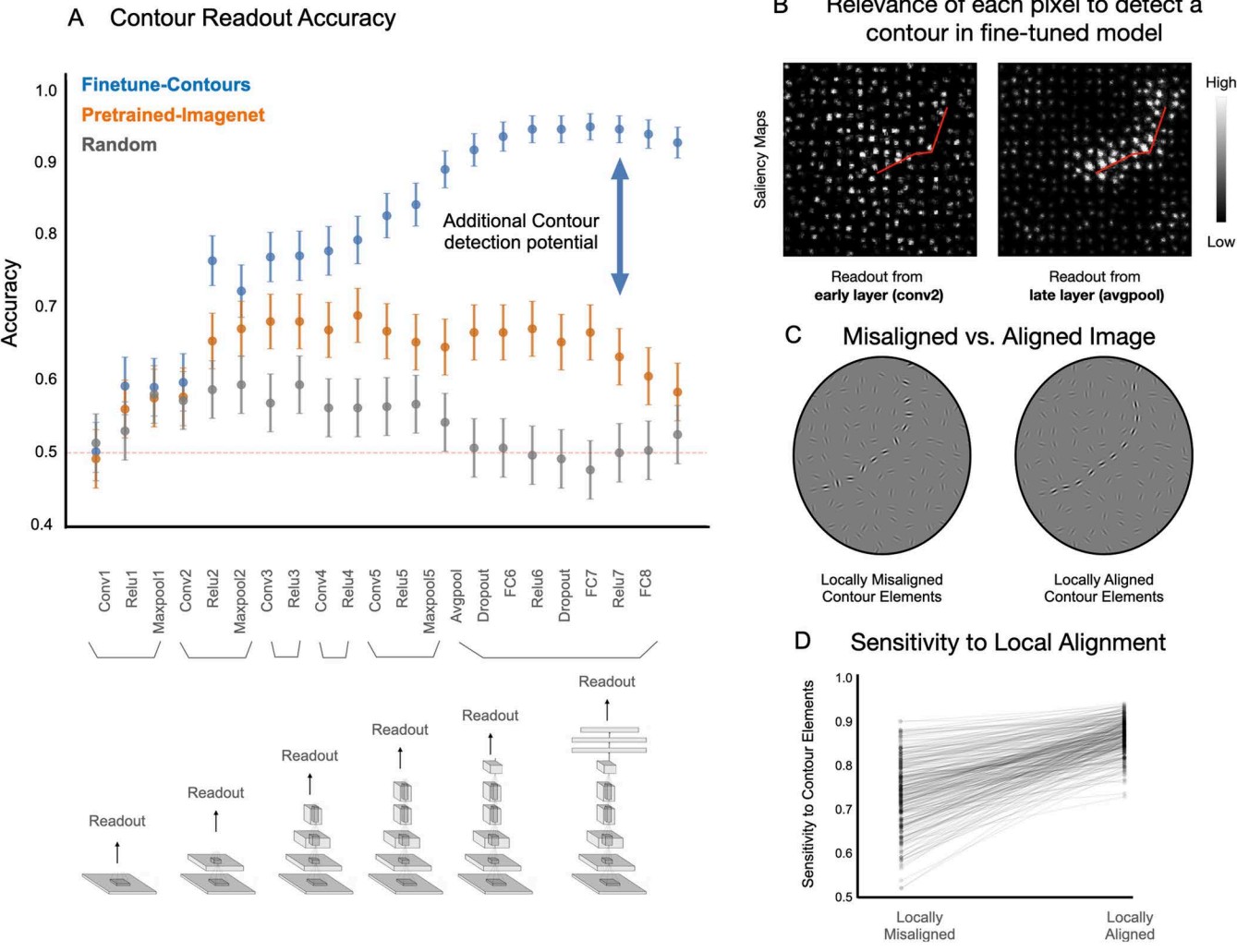

**Fig 2. Contour integration capacity in feedforward CNNs. (A)** Accuracy on contour detection for the held-out test set across different readout layers – gray indicating randomly initialized models, orange for models pretrained on Imagenet for object recognition, and blue for finetuned models. Error bars denote 95% confidence intervals for readout accuracy. **(B)** Saliency maps from two fine-tuned models, highlighting pixel relevance for detecting the contour within an example image; location of the contour is highlighted in red for illustrative purposes. **(C)** Example image pair with misaligned and aligned contour elements. Contour patches are highlighted for illustrative clarity. **(D)** Plot showing the fine-tuned model's sensitivity to elements making up the contour (for aligned display), or elements at the same locations in misaligned display. Each pair is connected via a gray line. Overall the plot depicts the sensitivity to local alignment of contour elements.

elements. Specifically, we used a feature visualization technique called Guided Backpropagation [63] to highlight the parts of the image that significantly contribute to the contour present/absent decision (see Methods for details on computing saliency maps). This method produces a "saliency map", where the relevance of each pixel to the final "contour-present" decision is highlighted.

In Fig 2B, we show the saliency maps over an example image where the location of the contour is highlighted with a red line for visualization purposes. We focus on read-out from two fine-tuned models, where the linear-classifier is attached to the "conv2" layer (early stage) and the "avg pool" layer (late-stage). The "avg pool" layer (the final stage of the convolutional backbone, before the fully-connected read-out layers), which is relatively late stage of the Alexnet model as

compared to the conv2 layer, shows higher contour detection accuracy (91.5% - avgpool layer; 59.6% - conv2 layer). It is clear that the avgpool model is leveraging more information from the contour elements along the path as compared to the background while successfully detecting the contour. In contrast, a linear read-out head attached to an earlier stage (conv2), shows little-to-no concentrated salience along the embedded contour. (See Methods and S2 Fig for quantitative results for all the layers). These results indicate that deeper layers in the feedforward architecture, but not shallow layers, have features that do in fact leverage the contour elements along the path to perform the contour detection task.

We next explore an even more nuanced potential concern. While the saliency maps depict sensitivity to contour elements, is successful performance actually arising from the *alignment* of these elements, and not just their locations, since the placement of contour elements is non-random? (see Methods; Stimulus Generation). To investigate this question, we conducted two separate analyses.

First, we compared the sensitivity scores for contour elements across image pairs (contour-present and contour-absent images) that differed only in whether the contour elements were aligned (forming a contour) or misaligned (no contour), maintaining the same locations for all elements as shown in Fig 2C. The model (fine-tuned end-to-end from the avgpool layer) showed higher sensitivity to contour elements when they were aligned, as depicted in Fig 2D. This result confirms that the model relies on fine-grained orientation information between contour elements to isolate the contour from the background. This pattern was also observed in models fine-tuned from other layers of Alexnet, with a greater emphasis on local alignment evident in models fine-tuned from deeper layers (refer to S2 Fig).

Second, we addressed the possibility that models use a heuristic solution to the contour-detection task that relies on local proximity plus orientation similarity to detect the contour, rather than relying on local continuity cues to detect the extended contour. To do so, we constructed new displays in which we displaced each contour element perpendicular to the path 0–8 pixels (see S3A–S3C Fig). This "alignment jitter" manipulation degrades the "good continuation" of the path while maintaining local proximity and orientation similarity between the contour elements. We tested the same fine-tuned model on these alignment-jittered stimuli, and find that the model performance systematically reduces as the alignment jitter increases (S3D Fig). Critically, performance drops to near chance levels at the +/- 8px offset, where local pairwise proximity and similarity are well-preserved, but local alignment is completely disrupted (the Gabor elements are ~16px). This finding provides strong evidence that the model's contour detection ability is not based on a simple heuristic of "proximity plus orientation similarity", and instead relies upon local continuity cues (good continuation) in order to identify which display has the extended contour.

Taken together, these results confirm the result that purely feedforward models have significant contour-detection capabilities, and can effectively support contour integration by stitching together elements based on local alignment cues.

## Progressive receptive field growth with layer depth underlies the model's ability to support contour integration

Another potentially deflationary account of this result is that success is trivial: any deep neural network can support contour integration because deep neural networks are universal function approximators. However, this property of universality only holds in the limit and does not necessarily apply to all networks (especially smaller ones). Consequently, it is possible for models to be architecturally incapable of successfully performing the contour detection task. We next aimed to design models exactly with this property in mind, in order to identify what aspect of the standard DCNN architecture was enabling successful contour detection.

Specifically, we hypothesized that the progressively increasing receptive field sizes of the Alexnet model architecture were critical for its capacity to detect contours. This hypothesis derived from the result that the read out from later stages, not the earlier stages, showed the strongest task performance. To examine this possibility, we designed a new model architecture, dubbed "PinholeNets", which are architecturally similar, but with a much more restricted field of view in their convolutional layers. This "cousin" model class thus allows us to investigate the impact of receptive field size progression on a model's ability to integrate extended contours, while holding constant as many other architectural aspects as we could.

We built four variants of PinholeNet models, here referred to as P11, P17, P31, and P33, named based on the receptive field size of units in the last (i.e., fifth) convolutional block. Hence the P11 model corresponds to an architecture whose unit in the fifth convolutional block would have a receptive field size of only 11 x 11 pixels. By comparison, the original Alexnet model has receptive field sizes of 195 x195px in conv5. Fig 3A shows the receptive field progression of these models across the layers, alongside an illustration of the RF size of a conv5 unit in the context of an example stimulus containing an extended contour. See Methods and S4A Fig for a detailed description of the PinholeNet architecture. It is critical to note that the number of kernels and hierarchical (non-linear) stages across all models is kept constant, and the only varying factor is the receptive field size of the units within the network.

Before testing whether these PinholeNets fail at contour integration, for comparative purposes it is important to first establish that these models are capable of learning useful representations at all, e.g., for classifying objects. Indeed, prior work has demonstrated that models with a restricted field of view can still be trained successfully on object recognition [62], revealing that very local cues in the ImageNet dataset are sufficient to support object classification [33,34]. We trained the PinholeNets on ImageNet for object recognition by pooling the final convolutional output across space, and then attaching the standard Alexnet classifier head (three fully connected layers). The terminal fully connected layer provided the 1000-way probability vector for ImageNet object categories.

We found that all PinholeNets attained a level of accuracy that is on par with the standard Alexnet model, with Fig 3B showing their top-1 performance on the held-out ImageNet validation set (P11 – 45.63%; P17 – 49.34%; P31 – 54.45%; P33 – 53.94%). Even though the standard Alexnet has the highest accuracy, it is just 10% more than the worst performing Pinholenet (in a task where chance accuracy is 1/1000, or 0.1%). Thus, despite their dramatically limited field of view in the convolutional backbone, PinholeNets are still capable of classifying objects at similar levels as their Alexnet cousin, and learn features that are useful for object recognition.

We next evaluated their capacity for contour detection after fine-tuning. When we fine-tune from the final (5th) convolutional layer, performance on the contour detection task increases as the model's receptive field size increases (Fig 3C, left), with all PinholeNets falling well short of the standard Alexnet model (converging at P11 – 52.33%; P17 – 57%; P31 – 61.83%; P33 – 70.66%; standard Alexnet 88.83% on the held-out test set). This result indicates that the smaller receptive fields hinder contour integration, even though the PinholeNets actually have higher-resolution output maps at this stage, and thus could have supported contour detection as well.

Are large receptive fields that encompass the entire image sufficient for contour detection? To address this question, we fine-tuned from PinholeNets units in the 2nd fully connected layer—where all units have full-field receptive fields (Fig 3C, right). We find that for the smallest two PinholeNets, there is little capacity to detect a contour by reading out over local features that can at most contain only one element (P11, P17). For the next two PinholeNets (P31, P33), there is improved contour detection accuracy when convolutional filters must span multiple elements. The standard Alexnet shows the highest contour detection accuracy, where the convolutional filters have access to multiple elements across the contour. This result indicates that full-field integration over sub-units that span multiple contour elements (as in P31, P33) improves contour detection, but that the gradually progressively increasing convolutional receptive fields support the most accurate contour detection.

Finally, we address the concern that maybe Alexnet's superior contour detection performance is just a byproduct of its slightly superior object recognition accuracy. To do so, we trained a standard Alexnet for half the normal number of epochs (50), at which point the model achieves a top-1 object recognition accuracy of 50.23%, aligning closely with the accuracy exhibited by the completely trained PinholeNets (between 45.63% and 53.94%). We then proceeded to finetune this half-trained standard AlexNet model on the contour detection task. We found that, even with a reduced accuracy in ImageNet object recognition, this model still exhibited superior performance in contour detection (89.33%; indicated by the black dotted line in S4C Fig) compared to the PinholeNets (between 52.33% and 70.66%).

Taken together, these results provide clear evidence that the architectural motif of progressively increasing receptive field sizes through a feed-forward processing hierarchy is particularly well-suited for undergirding contour detection.

## A. PinholeNet Architectures

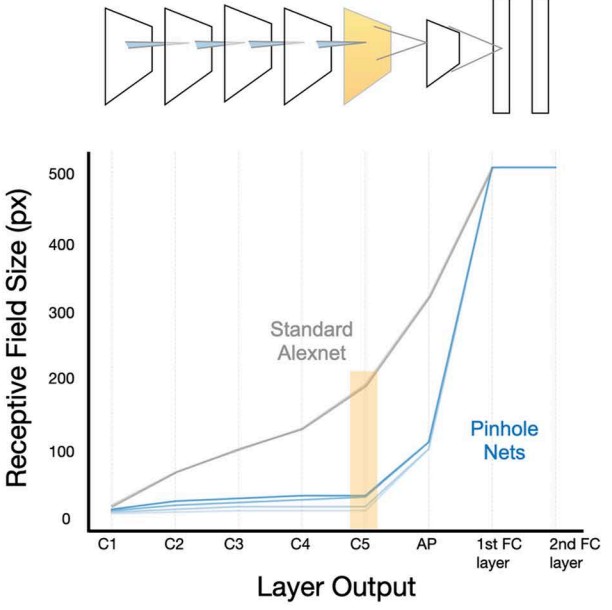

## Receptive Field Sizes
## Conv5 Block

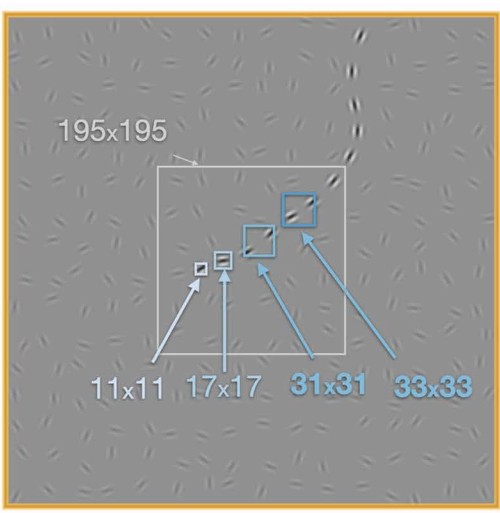

512 x 512 pixels

## B. Object Recognition Accuracy

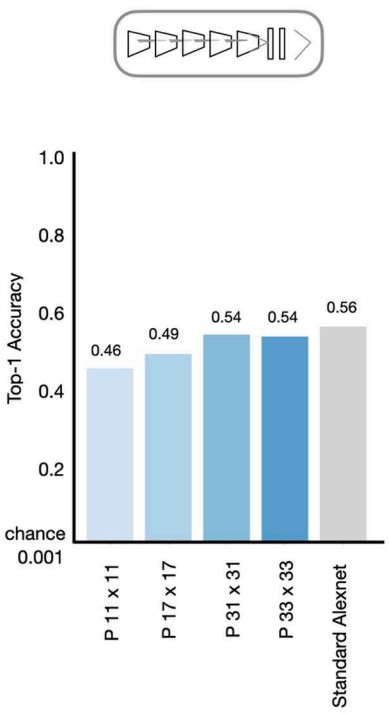

## C. Contour Readout Accuracy

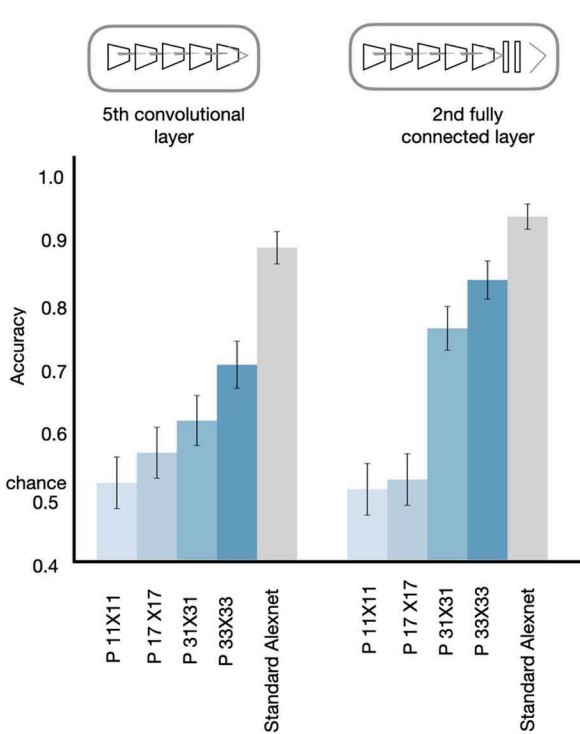

**Fig 3. Impact of receptive field size and progression on contour integration in feedforward models. (A)** left: shows the receptive field progression over the layers (blue lines), relative to the standard Alexnet model (gray); right: shows the size of the receptive fields of units in the 5th Convolutional block the final stage of the backbone before the fully-connected layers. **(B)** Top-1 object recognition accuracy on the ImageNet validation set for Pinho-leNet models with varying receptive field sizes (blue bars), as well as the standard Alexnet model (gray bar). **(C)** Contour detection accuracy for readout

from the 5<sup>th</sup> convolutional layer (left), and the 2<sup>nd</sup> fully-connected layer (right) in PinholeNets (blue bars) and the standard Alexnet model (grey bar) on the held-out test set. The error bars denote the 95% confidence intervals for readout accuracy.

## Human sensitivity to global path curvature

The previous results established the contour-integration capabilities of (certain) feedforward deep neural networks models, but it is unclear whether any of these models show human-like perceptual signatures of contour integration. To address this question, we examined whether the fine-tuned Alexnet models show a drop in contour detection accuracy as global curvature increases, which is a signature of human contour-integration. To do so, we had both humans and models perform a contour detection task for contours of various curvatures, replicating the human psychophysics paradigm introduced by [48]. Then, we compared contour detection performance on a trial-by-trial basis across humans and models. To do so, we also developed a principled method for comparing noisy human responses and noiseless model responses based on signal detection theory [64,65].

For this behavioral experiment, we generated a set of 2000 images (1000 contour present/absent controlled pairs), and varied the contour curvature across displays. The degree of curvature of the path in the contour-present image is described by the angle between successive contour elements along the path (± β). This curvature level was fixed for a path in an image (15°, 30°, 45°, 60°, or 75°), with smaller β values resulting in displays with straighter contours, and larger β values leading to curvier and more irregular contours (see Fig 4A). For the contour-absent image, contour elements were positioned at exactly the same locations but were randomly oriented. The background elements were identical for the contour-present and contour-absent images in a given present/absent pair (see Methods section for additional details on stimulus generation), but this alignment was made inconspicuous by randomly rotating the two images (0, 90, 180, 270 deg), and ensuring they were offset by at least 90deg. The dataset was randomly split into five sets of 200, with 40 pairs for each beta condition in each set, and each participant was shown trials from one of these five sets selected at random.

Participants performed a 2-interval forced-choice (2-IFC) contour detection task, where they were presented with two displays (one with the contour-present image and another with the corresponding contour-absent image) and had to identify which one contained an extended contour. A typical trial is depicted in Fig 4B. Over the course of the experiment, participants encountered 200 trials with contours at all levels of curvature (with trials counterbalanced across ten conditions: five distinct β values, presented in either the first or second interval of the trial). Different trial-types were uniformly distributed over the full experiment. See the Methods section for behavioral experimental design. To ensure the reliability of our results, we excluded participants whose performance was beyond 2.5 standard deviations of the group mean accuracy or whose individual performance trends across all β conditions did not correlate well with the group trend (see exclusion criteria in Methods and S5A Fig). Our final analysis was based on a refined pool of 78 participants who met these exclusion criteria (S5C Fig), but all trends were replicated with the full dataset (S14 Fig).

We plot the mean accuracy across the five curvature levels (β conditions) for these participants in Fig 4C. The average results across curvature conditions shows that participants were able to accurately detect relatively straight contours, and that the task becomes increasingly difficult as the beta value increases (Fig 4C).

However, there is additional richness in these data revealed by plotting the performance for individual trials (measured as percent correct across participants) as shown in Fig 4D. Here, we find substantial variation across the trials within displays of the same curvature level (see S6 Fig for examples). And, these differences are highly consistent across participants, with the split half reliability across participants at r = 0.65 and the Spearman-Brown adjusted estimate of reliability at 0.786, 95% CI [0.777 to 0.794]. Critically, the reliability of these trends enables us to compare the strength of contour representations at the level of individual trials (i.e., each image pair).

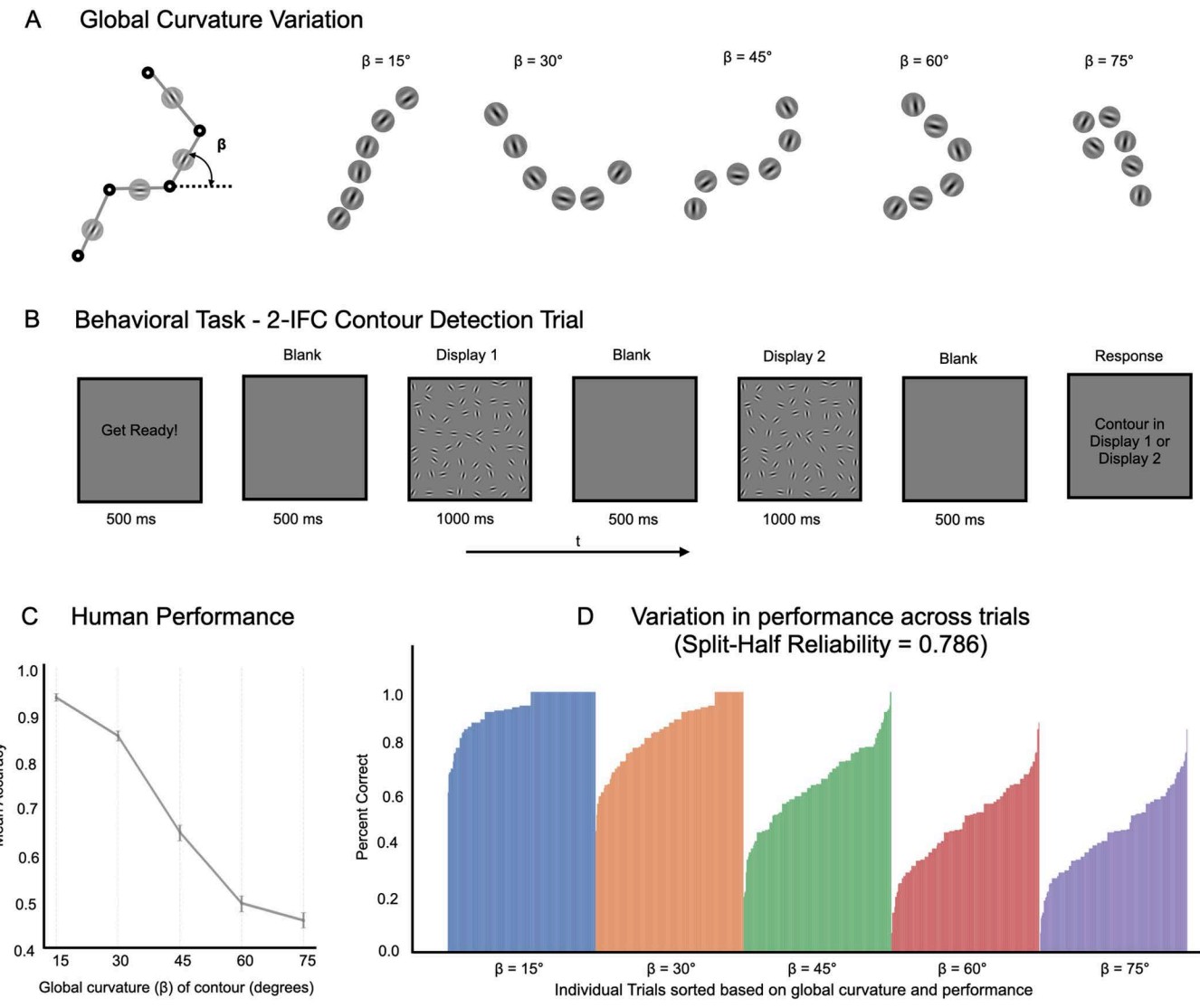

**Fig 4. Human sensitivity to global curvature. (A)** Variation in global curvature (β) across a range of contour stimuli used in the study, with β values set at 15°, 30°, 45°, 60°, and 75°, demonstrating straighter to more curved contours. **(B)** Sequence of a 2-IFC contour detection trial where participants identify the display containing the contour. **(C)** Mean accuracy of participants for contour detection across varying β conditions, with error bars representing 95% confidence intervals of the mean accuracy bootstrapped across participants. **(D)** Bar graph showing the variability in human performance across individual trials within each β condition.

## Estimating decision strength in deterministic models

One challenge in comparing human and model decisions on a trial-by-trial basis is that these models, like most deep neural network models, are deterministic: they will give the same response to an image every time it is presented, and thus their decisions can only yield a binary outcome (correct or incorrect), which can mask the fact that some contours are easier to detect than others.

To address this challenge, we use a signal-detection framework to compare model and human decisions [65]. The idea here is to estimate, separately for both models and humans, what the contour signal strength is for each individual trial.

For models, the estimate of the contour signal strength was derived by comparing the activation on the "contour present" node for a contour-present image and its matched contour-absent image from the stimulus dataset used in the behavioral experiment. The stronger the "contour present" response is for a contour-present image relative to contour-absent image, the stronger the "contour signal" for that specific image-pair. This is estimated in equation 1 (see signal detection analysis in Methods and S7 Fig) where *response* is measured in the contour-present node for each image pair.

$$\frac{response_{contour-present\ image} - response_{contour-absent\ image}}{\sqrt{2}} \tag{1}$$

For humans, percent correct on a 2AFC task can be considered a measure of signal strength [64]. Thus, of principal interest is whether the strength of contour signals in the model predict human percent correct for the same image pair.

### Fine-tuning to detect low curvatures gives strongest alignment with human behavioral data

Do models fine-tuned to perform contour detection have the same curvature perceptual signatures as humans? Recall the human behavioral data indicated that detecting contours with high global curvature is challenging. We first considered a model that was fine-tuned to perform contour detection over all curvatures spanning a broad range (global curvature parameter β varying broadly as shown in Fig 5A, with the classifier attached to the avgpool layer). Importantly, by doing so, we are testing the capacity of the model to perform contour detection across a broad range of contour curvatures, but this does not guarantee that it will be equally performant across the full range—it's possible that this model will also show the same characteristic dropoff in detection accuracy with curvature that people do. However, at a trial-by-trial level, this model shows only a weak correlation with human percent correct across the 1000 trials/image-pairs (Fig 5B red scatter plot; Pearson's r = 0.1907, 95% CI [0.13,0.35], p < 0.001). On average, this model fails to capture the characteristic dropoff in contour detection performance as curvature increases (Fig 5C, line plot, red vs. dark gray lines). Instead, it was capable of detecting curvatures better than humans, particularly for the more extreme curvatures. This correspondence failure also occurs in models that were fine-tuned from other layers of Alexnet (S8 Fig).

What would it take for the model to show a more human-like pattern of contour detection? The human behavioral data indicate that detecting contours with high global curvature is challenging, and is best for straighter lines, with a gradual fall off with curvature. Thus, we hypothesized that if we limit the model's fine-tuning training to only straight line contours, we might naturally recapitulate this human-like characteristic fall off, and potentially account for the trial-by-trial variation as well. In the following analyses we compared a set of models, each with fine-tuning to a single level of curvature in the displays, from straight to gradually curved (0° to 30°). Then, we tested those models on the same displays used in the psychophysics experiment. It's important to note that we are not fitting the model to human data by training on the human data in any way. Instead, we are exploring a range of possible hypothesis about what curvatures the model system needs to be fine-tuned to, and seeing which (if any) of our fine-tuned models show emergent human-like patterns of decisions across the 1000 displays of the experiment.

Starting with a baseline model trained on Imagenet, we created 16 new model variations that were fine-tuned to detect contours, each at a single curvature (β) level, from 0° (perfectly straight) to 30°, in 2° increments. Note in all these models, the contour classifier was attached to the avgpool layer. Fig 5D plots how strongly each of these fine-tuned models correlated with human trial-by-trial responses. Contrary to our expectations that the model trained on purely straight edges would best recapitulate the human data, we instead find that models fine-tuned to detect curvatures around β = 20° showed the highest agreement (r = 0.77, 95% CI [0.7413, 0.7923], p < 0.001). Indeed, the level of correspondence approached the human data noise ceiling (r = 0.786). Fig 5E shows the trial-by-trial green scatter plot, with the 20°-fine-tuned model's estimate of contour signal strength on the x-axis, and the human's percent correct for that the trial on the y-axis. Fig 5F, shows that, aggregating across trials with the same curvature, the model shows the same characteristic dropoff in performance as in the human data. Finally, we also find that fine-tuning on higher curvatures (45°, 60°, or 75°)

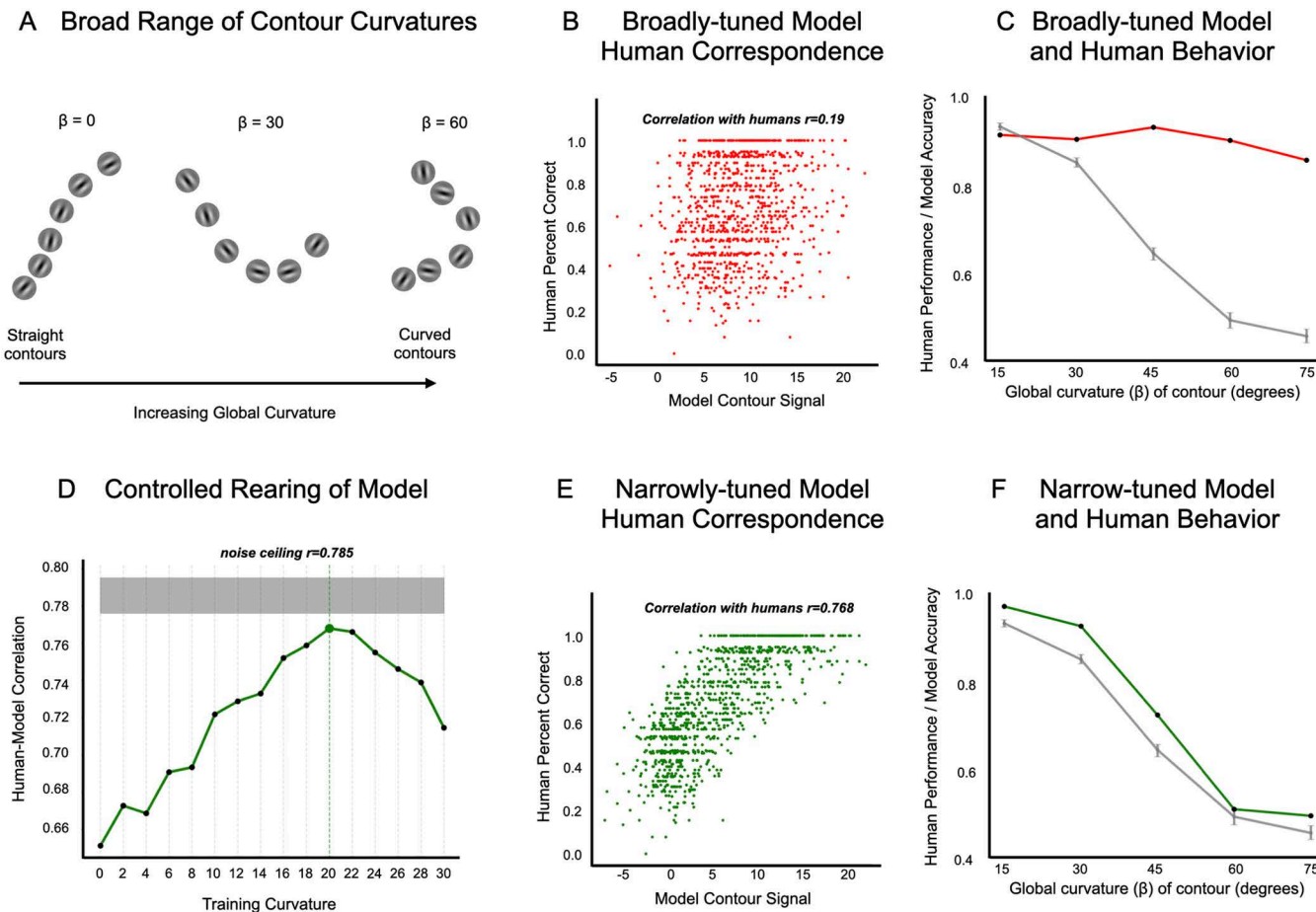

**Fig 5. Model and human behavioral correspondence for contour integration. (A)** Contour stimuli containing global curvatures (β) spanning a broad range. **(B)** Scatter-plot depicting the correlation between the broadly-tuned model's contour signal strength and human percent correct across trials, showing weak correspondence (Pearson's r = 0.1907). **(C)** Line plot illustrating the broadly-tuned model's performance against human performance for different global curvature levels, highlighting the model's insensitivity to increasing curvature. The broadly-tuned model's performance is shown in red and human performance is shown in grey. **(D)** Line plot illustrating the correlation of models, that were trained on curvatures within a specific narrow range (resulting in narrowly-tuned models), with humans, peaking at β = 20° and approaching noise ceiling (r = 0.785). **(E)** Scatter-plot depicting the correlation between the narrowly-tuned (at 20°) model's contour signal strength and human percent correct across trials, showing strong correspondence (Pearson's r = 0.768). **(F)** Line plot illustrating the narrowly-tuned model's (at 20°) performance against human performance for different global curvature levels, highlighting the human-like sensitivity to curvature. The narrowly-tuned model's performance is shown in green and human performance is shown in grey.

results in lower correlations with humans (r = 0.2, r = -0.49, r = -0.63), as detailed in S9 Fig. These results also generalize to linear-classifier probes trained on other layers (see S10 Fig), consistently showing peak human correspondence when fine-tuned with contours whose curvature is around 20 degrees (± 4 degrees).

Note that this emergent fit is by no means guaranteed. It quite reasonably could have been the case that fine-tuning a model on only one curvature level would show little generalization to other curvatures, with a steep fall off in accuracy for out-of-distribution curvatures (as is often the case with deep net fine-tuning). For example, the model trained on 20° curvature might show poor contour detection accuracy at 0°, 10°, 15°, and 25° etc. In this case we might have had to explore more complex fine-tuning protocols that include exposure to a range of curvatures, perhaps with different frequencies for each level. However, our experiment yielded a relatively simple and strong finding: fine-tuning solely on 20 degrees of contour curvature ends up capturing human-like contour detection variation across all images of varying curvatures, at near

noise-ceiling of the highly reliably human data we collected. And, as result, the model also demonstrates the right slope in the average performance across different curvature levels (See S11 Fig for a plot of the asymmetric generalization performance of the 20°-fine-tuned model accuracy over a more granular set of curvature levels).

Broadly, this emergent match for the 20° fine-tuned model suggests that human contour integration mechanisms might actually be optimized for this relatively narrow band of gradually curving contours, where feedforward CNNs can emulate these signatures of human perception when their contour-integration mechanisms are biased towards encoding relatively gradual curves.

### Assessing emergent contour detection capabilities in other network architectures

In an auxiliary analysis, we also explored how other standard and potentially more powerful vision model architectures would fare on the contour detection task. Specifically we focused ResNet-50 [66] and a Vision Transformer [67–69], testing whether these pre-trained models (without additional fine-tuning) exhibit better emergent contour detection accuracy than the standard Alexnet model we used. ResNet-50 was selected as the current industry standard deep CNN architecture, with more layers and residual (skip forward) connections, and has superior ImageNet performance than the classic Alexnet architecture. The Vision Transformer was chosen because it has within-layer self-attention mechanisms, learning how to associate information across different patches of the image, which could facilitate lateral information flow and potentially enhance contour detection.

To carry out these analyses, we kept the backbone features fixed, and trained linear probes to test whether contour presence could be read-out from each of the hierarchical layer blocks (relu blocks of the Resnet50; and the class token representation for vision transformer). To train the linear readouts, we showed each model contours with curvatures spanning a broad range, as we did in our first analysis with the Alexnet architecture, to assess the degree to which detecting an extended contour is possible. For vision transformers, note that we resized our contour displays to fit the vision transformer front end size of 224 x 224, which meant that each input patch had access to an individual gabor element from the display, plus sometimes a partial view of another gabor element depending on the jitter. (see Methods for additional details).

Do these models, in their pre-trained state, inherently support the detection of extended contours based on their learned representations? Performance accuracy for contour detection across intermediate layers of these models is shown in S12 Fig. ResNet-50's contour detection performance is comparable to that of the pre-trained AlexNet model, with a maximum accuracy of approximately 69%, but remains far below the fine-tuned AlexNet model shown in blue. This result suggests that even the modern DCNN pre-trained models lack the strong inductive biases required to integrate local information for detecting extended contours. The Vision Transformer, starting at chance level in the earlier layers, shows a gradual but marginal increase in performance through the later transformer blocks, reaching a maximum accuracy of 58%. This finding is important because, despite containing explicit associative processing between image patches, pre-trained Vision Transformers did not exhibit a significantly enhanced capacity for contour detection. This limited capacity could also be attributed to factors such as insufficient training data or suboptimal hyperparameters, like the optimizer or learning rate scheduler, while training the linear readouts. However, since this experiment did not involve fine-tuning, the same hyperparameters were used for linear readouts across all models, simply testing whether relevant information for contour detection was present in the intermediate layers of each network. Taken together, these findings suggest that, while modern pre-trained architectures such as ResNet-50 and Vision Transformers exhibit some capacity for contour detection, they still lack the necessary inductive biases to show emergent contour integration capability.

### Assessing the generality of a feedforward architecture's capacity in other grouping phenomenon

Although our findings demonstrate the sufficiency of feedforward mechanisms for contour integration, it remains an open question whether such architectures are sufficient for other forms of perceptual grouping. For example, Doerig et al. [42,45] have argued that the phenomenon of visual "uncrowding" cannot be captured by purely feedforward networks. In

this phenomenon, performance on a vernier acuity task, which is impaired by a single flanking object (crowding), paradoxically improves with the addition of more flankers (Fig 6A). Because this effect depends on the global configuration of the flankers, it is considered a signature of a complex grouping mechanism. While previous work tested networks with a frozen backbone to make claims about architectural capacity, it left untested whether fine-tuning a feedforward architecture could reveal a capacity for uncrowding.

To distinguish between these possibilities, we generally followed a similar training and testing procedure as used in prior work (see Methods for stimulus parameters). Specifically, we trained a VGG19 model—a purely feedforward architecture—on a vernier discrimination task, using training displays where the vernier and flanker configurations were presented simultaneously but did not overlap (Fig 6B, left). The model was then tested out-of-distribution on displays where the vernier was centered within the flanker configurations (Fig 6B, right). Critically, we compared two scenarios: one where the pretrained VGG19 backbone (pretrained on object recognition over ImageNet) was kept frozen, and one where the backbone was fine-tuned end-to-end. In both cases, a linear readout head was attached to the final average pooling

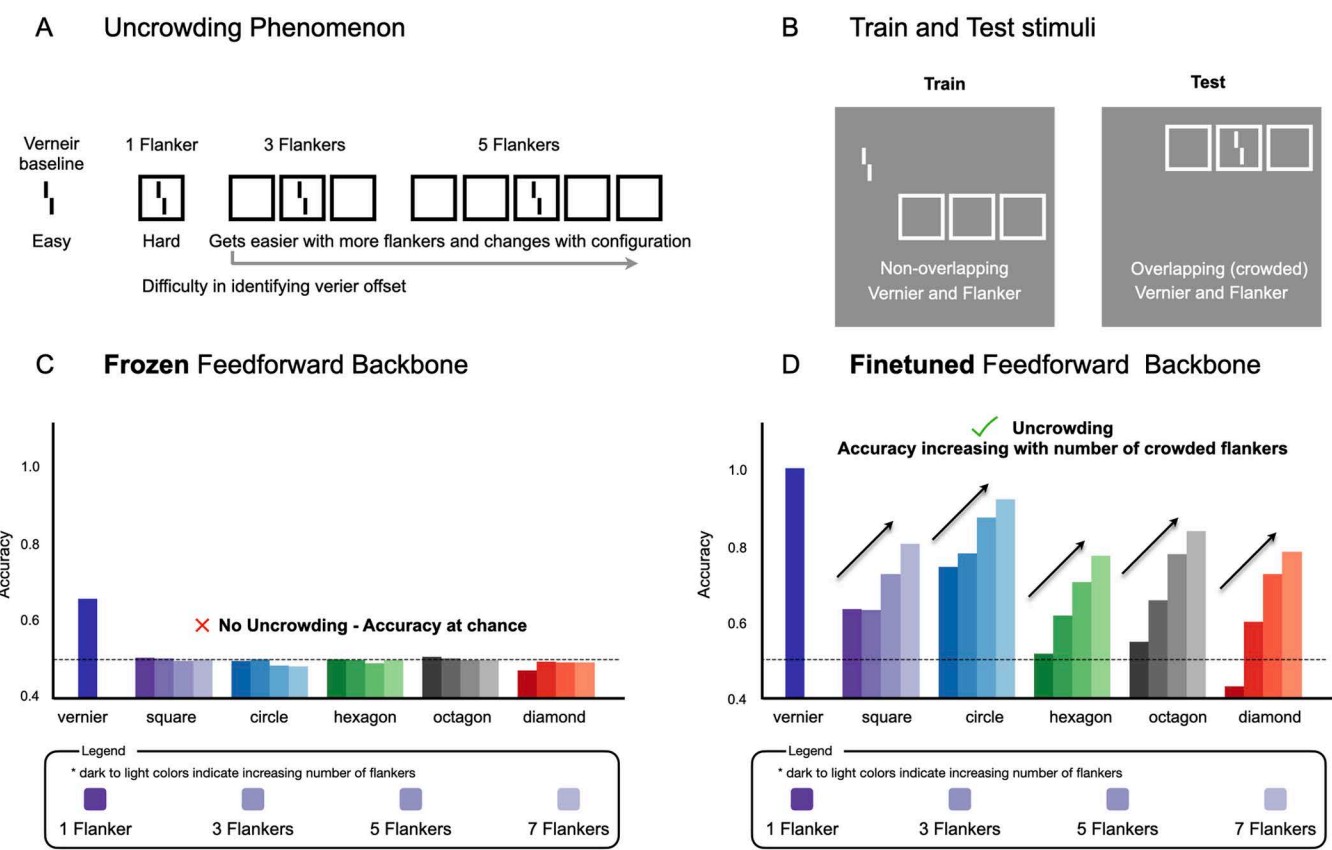

**Fig 6. Fine-tuning a purely feedforward network reveals a capacity for visual uncrowding. (A)** A schematic of the uncrowding phenomenon. Identifying the offset of a vernier target is easy when presented in isolation (baseline), becomes difficult when surrounded by a single flanker (crowding), and becomes easier again as more identical flankers are added to the configuration (uncrowding). **(B)** The out-of-distribution training and testing paradigm. Models were trained on non-overlapping stimuli, where the vernier and flanker configurations appeared in the same image but were spatially separate and tested on overlapping (crowded) stimuli, where the vernier was centered within the flanker configuration. **(C)** Performance of a VGG19 architecture with a frozen, pretrained backbone. While the model can identify the vernier in isolation, its accuracy drops to chance level for all crowded conditions, regardless of the number of flankers. The model fails to exhibit uncrowding, consistent with prior reports on the limits of pretrained feedforward architectures. **(D)** Performance of a VGG19 network with a fine-tuned backbone. Accuracy is high for the isolated vernier, drops with a single flanker, and then systematically increases as more flankers are added. The results shown are from the model with the clearest emergent uncrowding.

layer (see Methods). The frozen backbone condition served as a conceptual replication of prior work, while the fine-tuning condition tested whether the feedforward architecture can, in principle, perform the task and emergently capture the uncrowding phenomenon.

With the frozen backbone, the VGG model performs the vernier task above chance in the unflanked condition, but shows no ability to do so under any of the crowding conditions; performance simply drops and remains at chance (Fig 6C). This result is consistent with previous reports [42,45] that pretrained feedforward CNNs fail to exhibit uncrowding. However, critically, when fine-tuned (Fig 6D), the same model not only performs the task successfully but also qualitatively demonstrates emergent uncrowding: it shows high accuracy for the isolated vernier, reduced accuracy when surrounded by one flanker, and progressively improving accuracy with an increasing number of surrounding flankers. Thus, purely feed-forward networks can show emergent "uncrowding" signatures, without further architectural mechanisms.

It is important to note that while we demonstrate the principle of sufficiency here, the results were not universally this clear across all architectural variations. For example, when we first used an AlexNet architecture on this task, we did not replicate the same pattern of results. The AlexNet model was in fact never able to learn vernier judgments in crowded conditions, even when trained directly on the overlapping stimuli themselves. One reason for this failure could be the large stride and filter sizes in the initial convolutional layer of AlexNet, which may prevent the architecture from resolving the fine-grained vernier stimulus with a nearby flanker. In this regard, AlexNet could be considered architecturally incapable of uncrowding, but this appears to be more about limitations in processing the input rather than a universal requirement for recurrence, given the VGG19 results. We in fact turned to the VGG model precisely because its architecture, with smaller, stacked convolutional filters and reduced stride, allows for a finer-grain sampling of the visual input. A second qualifier is that even within the successful VGG19 model class, we observed variability across training runs (S13 Fig). In a set of eight training runs, we found that typically one in eight models failed to converge meaningfully. Of the remaining successful models, a subset (approximately 2 of the 7) showed the clear, parametric uncrowding effect, while others performed failed to show a systematic generalization gradient. The existence of such variability across identical models trained on the same data serves as a valuable cautionary tale for drawing strong conclusions from single model instances [70]. However, for the purposes here, the existence proof that at least *some* fine-tuned VGG models are capable of showing emergent uncrowding is sufficient.

In sum, while fine-tuning was required to unlock the feedforward architecture's capacity for emergent uncrowding, these results show that such a network has no inherent architectural barrier that prevents it from demonstrating the uncrowding phenomenon, and that under the right conditions these capacities can be revealed. As with our contour integration results, this still leaves open the question of what is missing from standard pre-training (e.g., on ImageNet) that prevents models from naturally succeeding at these tasks without task-specific fine-tuning. One possibility is that the psychophysical experimental stimuli are typically out-of-distribution compared to natural images. Allowing the models to fine-tune their weights may reveal these feed-forward grouping capacities because the model can better represent and distinguish these stimuli. On this account, a solution to the domain-shift problem alone may be sufficient for ImageNet pre-trained models to show emergent human-like grouping capacities, even without fine-tuning on the psychophysical stimuli and task. Understanding the interplay between the task-specific training and the stimuli data distribution shifts in revealing this perceptual-organization capacities of networks is an open question.

## Discussion

Deep neural networks provide extensive control over factors crucial to learning visual representations, including the visual diet, neural architecture, learning objectives, and regularization pressures, and are particularly well-suited to making computational plausibility arguments — i.e., understanding what factors are necessary and sufficient for different kinds of representations to be learned. As such, these models provide a powerful experimental platform for exploring theories of visual processing, including investigating the mechanisms underlying perceptual grouping and the formation of high-level holistic

representations. In the present study, we focused on the contour integration capabilities of deep convolutional neural networks, and found that these purely feedforward systems can support human-like contour detection, provided they have progressively-increasing receptive fields, and a bias toward relatively smooth/gradually-curved contours. Moreover, we found that these models performed contour detection using local alignment of features to detect chains of local elements and integrate them into a higher order structure — a contour. Thus, despite being supported by units with large receptive fields, the integration still operates over fine-grained local details, highlighting that large-receptive fields need-not encode coarse representations. Together with our proof-of-concept demonstration of visual uncrowding in feed-forward models, these results suggest that feedforward hierarchies may support a broader class of perceptual grouping computations than previously assumed [42,45].

It is important to emphasize that these results do not contradict or undermine prior work suggesting that lateral connections and feedback play an important role in contour integration [48,54,56,61,71]. Instead, they show that lateral connections and feedback mechanisms are not necessary for contour integration, that purely feedforward processing along a hierarchy with progressively increasing receptive fields is sufficient. As an additional point of clarification, these DCNN models do use backpropagation during training to adjust their weights, which might be considered as a kind of feedback. However, here we note that, once trained, the actual processing of the display images, and flow of information en route to a contour detection decision remains strictly feedforward. This means that, once the network is trained, all computations proceed in one direction, from earlier to later layers, without any feedback occurring between them, and this process is sufficient to support successful contour detection in these models.

At the same time, we find that CNN models pre-trained on ImageNet classification alone was not sufficient; these models did not show an emergent match with human behavior. And, this pattern of data is in broad agreement with prior work [39–41]. Instead, our analyses revealed that fine-tuning with an inductive bias towards gradual curvature was required to see more human-like contour detection decisions. In particular, models fine-tuned on gradual curvatures (~20 degrees) were critical for showing emergent trial-level correspondence with human judgements; while models fine-tuned with contours <20° or >20° did not show as clear an emergent match. However, it is important to clarify that our findings do not suggest that the human visual system acquires contour integration abilities through the same task-specific fine-tuning process used in these deep neural networks (DNNs). Instead, these results provide a clear direction for future research — to discover new inductive biases that can be built into DNN models so that human-like contour detection emerges without any fine-tuning to the contour task. For example, perhaps more human-like contour detection might naturally emerge in in models trained on even more general visual objective (e.g., self-supervised learning or predictive coding over more 3-dimensional object views). Thus, our primary claim here is that it is possible for a purely feed-forward hierarchy to perform contour integration in a human-like fashion, but that the models are lacking some key ingredients to show a fully *emergent behavior* that aligns with humans, without any task-based fine-tuning.

## A synergistic view of feed-forward, lateral, & recurrent mechanisms

We find evidence that a purely feedforward system can implement human-like contour detection, implementing contour-integration mechanisms that are sensitive to local-alignment cues and stitching them together across a hierarchy of progressively growing receptive fields. We believe this is an important discovery because the role of the feedforward hierarchy in contour integration and perceptual grouping has been relatively understudied in prior work, which has assumed that relatively low-level mechanisms perform contour integration by amplifying responses to local elements that contain colinear neighboring elements (see **Fig 7A** for illustration). For instance, models with lateral/top-down connections and/or recurrent units can capture human perception of contours [48,71], as well as neurophysiological aspects of contour detection [54,56,61], including within natural images [57–60]. However, one conceptual issue with early-stage contour-integration mechanisms is that they serve primarily to boost responses to features that are part of a contour, but those responses are still localized and ambiguous — e.g., a particular neuron sensitive to a particular orientation can give a

strong response due to a number of factors, from increased contrast of the local features, better alignment to the preferred orientation, and/or the presence of colinear elements outside of the classical feed-forward receptive field. Integration across such local units is necessary to encode properties of the full set of contour elements (i.e., "the contour").

Our current feedforward model thus suggests an alternative, feedforward mechanism for contour integration, where binding and grouping are interwoven within a multi-level hierarchical framework, driven primarily by feed-forward mechanisms which represent the "integrated whole" at the top of the hierarchy. Each convolutional layer in this framework represents similarities and differences over space, with subsequent layers incorporating aspects of non-transitive grouping [72]. This feedforward computation starts by isolating potential candidates of colinear elements over smaller receptive

## A    Early-stage contour representation via lateral computation

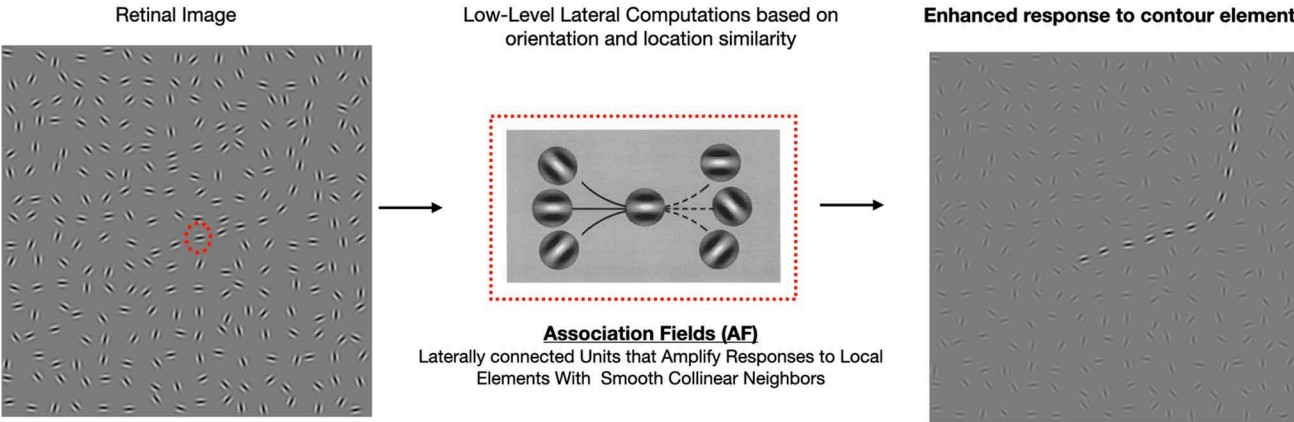

## B    Mid-stage contour representation via feedforward computation

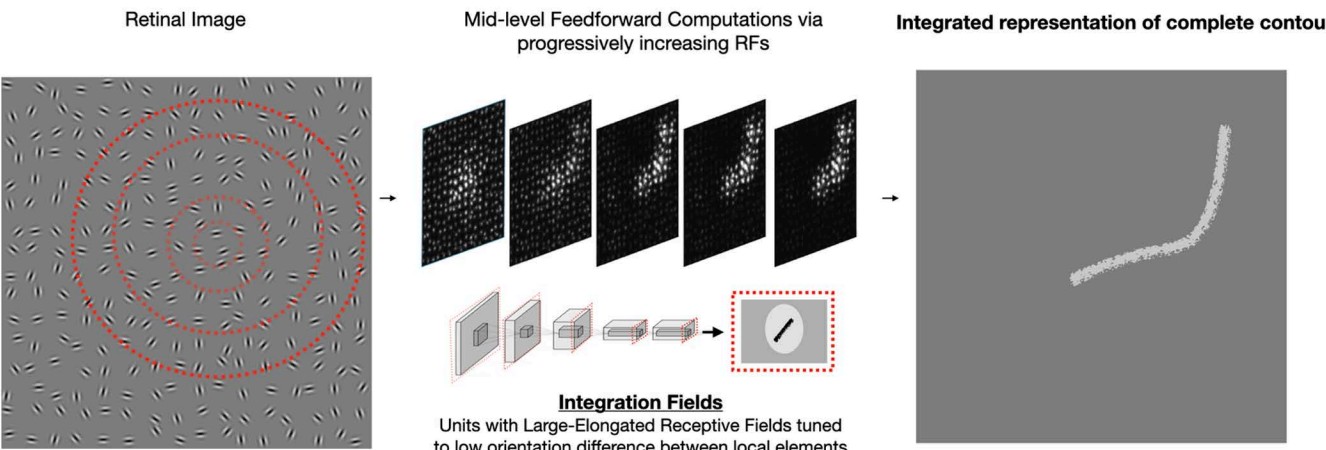

**Fig 7. Computational mechanisms underlying human contour integration. (A)** Low-level computations amplify responses to local elements that are part of a contour in the retinal image. This is facilitated by lateral connections between units with collinear tunings conceptualized as Association Fields **(B)** Mid-level feedforward computations focus on identifying potential candidates for an extended contour in the retinal image. This is facilitated by units with progressively increasing receptive fields (RFs) that are tuned to low orientation differences, allowing for the integration of local features into coherent extended contours.

fields, and then by processing over larger and larger receptive fields, successively chips away accidental collinear elements until it narrows down on the complete contour (see Fig 7B for illustration). This same feedforward mechanism may also underlie other forms of perceptual grouping beyond contour integration — such as uncrowding — which we found to emerge after fine-tuning on the vernier task. The fact that the model emergently shows reduced interference with increasing numbers of flankers suggests that the feedforward hierarchy is capable of representing not just the presence of contours, but also the global context in which contours and shapes are embedded.

Taking prior work and the current results into account, we propose an integrated framework in which the early lateral/recurrent operations serve to amplify local responses to elements that are likely components of extended contours, but that feedforward processing plays a primary role in integrating these local features into a complete representation of the extended contour. This raises the possibility that later stages of human visual processing, such as higher-order areas (e.g., IT cortex), may play a more substantial role in contour integration than traditionally assumed, shifting the focus beyond early visual interactions in V1 and V2 that have historically dominated contour processing models. On this account, it is the tuning properties of these high-level integration units (and not the previously proposed early-stage mechanisms) that underlies human contour perception by providing an explicit representation of the integrated "whole" beyond the simultaneous localized representation of its elemental "parts" — i.e., making the whole greater than the sum of its parts [11].

### Limitations of the approach

We examined contour integration using a relatively constrained stimulus set, specifically Gabor elements with specific parameters and contours comprised of a fixed number of local elements. This approach enabled us to isolate contour-integration mechanisms and provided a high-powered comparison with human psychophysical data. A promising future direction would be to explore how the integration units respond to systematic variations in other stimulus dimensions, such as frequency, scale, density, phase and symmetry [73–76]. Similarly, uncrowding paradigms that systematically manipulate global flanker configuration while holding local target properties constant could provide a rigorous benchmark for testing the extent to which different models capture global spatial dependencies critical for human-like perceptual organization. In our view, these steps would be most fruitful following advances in deep neural networks that lead to emergent alignment with humans, as noted above and discussed in detail below.

Furthermore, our investigation also focused on contour perception in two-dimensional images, but research by [77] suggests that the visual system also engages in contour integration across different depth planes. Exploring these dynamics with deep neural network frameworks, particularly those that are depth-aware [78,79], could offer new insights into how the visual system transitions from an image-centric representation to one that is more object-centric.

Finally, in the current study we used a linear classifier probe trained directly on contour detection to assess the contour-integration capacity for different model layers and to enable direct comparisons between model and human performance. An alternative approach would be to assess the tuning properties of individual units within each model layer, and to draw parallels between those tuning properties and neurophysiology or behavioral data. For instance, future research could isolate artificial units sensitive to the alignment of local elements and test whether these units exhibit a preference for gradual curvatures in a way that parallels human contour detection sensitivity. We see this approach as complementary to the work presented here, and provide resources for generating stimuli that can be useful for such *artiphysiology* research.

### Towards emergent human-like behavior

An ambitious goal for vision science is to develop a deep learning framework that inherently exhibits human-like perceptual organization through exposure to natural visual diets, biologically inspired architectures and task objectives. Such a system would naturally demonstrate human-like perceptual phenomena, including human-like contour integration,

when subjected to the same psychophysical tasks as humans, and would not have to be fine-tuned on the probe stimuli. Although training with datasets like Imagenet can enhance these models' contour integration abilities compared to their untrained counterparts, it remains unclear what additional inductive biases are needed for these models to show, without further fine-tuning, the characteristic human degradation with increased contour curvature.

Why are humans more sensitive to relatively straight contours? One possibility is that human perception is optimized for curvatures that are most prevalent in their visual diet, and that relatively straight contours are more prevalent in natural images. Indeed, research indicates a bias in natural image distributions toward relatively straight edges, often with orientation differences peaking near 20 degrees between adjacent elements [46,47]—a value that is intriguingly comparable to the inductive bias that provided the optimal alignment between our fine-tuned models and human contour detection accuracy. On this account, networks that learn over images with this natural image bias may inherently become tuned to relatively straight edges. While ImageNet contains about 1.3 million images, it is unclear whether they are adequately representative of natural image statistics needed to induce this bias. Training networks on more extensive and varied naturalistic datasets [80] could potentially enable them to develop more nuanced representations of extended contours, more closely approximating human sensitivity to curvature variations. Additionally, using ego-centric datasets like the developmentally realistic SAY-Cam [81] or EGO4D dataset [82], or high-resolution input images that can seemingly increase shape reliance without data augmentation [83] may further enhance the network's ability to learn rich, realistic visual representations.

Nonetheless, we speculate that statistical regularities in the natural visual diet must be combined with strong pressures to form structured representations for models to demonstrate human-like contour integration. Standard computer vision tasks, such as object recognition, or self-supervised objectives, do not appear to require such structured, shape-based representations [84,85], resulting in deep neural network models that prioritize local color and texture information over extended contours and shapes [33,34,85], and generally lack sensitivity to the spatial configurations between object parts [36,38]. Indeed, the BagNet models are a direct showcase of these purely local strategies, as they have only small, local receptive fields and can still achieve near equivalent ImageNet accuracy as models with convolutional hierarchically increasing receptive fields [62]. A tendency for models to learn local textural strategies may also help explain why uncrowding effects do not emerge in standard pretrained models—such effects depend on global spatial configurations that may simply not typically represented or required in conventional classification tasks. Like contour integration, achieving *emergent* uncrowding from pre-training alone may require a combination of richer visual diets and stronger architectural or objective constraints that promote global grouping sensitivity. While some efforts have been made to enhance shape-representation in deep neural network models [34], improvements in shape-bias have not been accompanied by high-quality, human-like shape-representations [86]. We propose that stronger objective pressures to form geometric shape representations [87] are needed for the emergence of human-like contour integration in these models.

In addition to changes in visual diet or task objective, it is also possible that these networks require modified architectures that are better suited towards contour integration. While out-of-the-box pretrained Vision Transformer did not exhibit enhanced contour integration capacity compared to DCNNs, its self-attention mechanism may still hold promise for future explorations of contour processing, particularly if combined with other mechanisms. One promising direction can be to incorporate ganglion and LGN-like processing in the early layers of the network. These cells are adept at encoding crucial edge and contrast information via their distinct center-surround receptive fields [88]. Emulating these characteristics can potentially enhance nuanced detection of orientation and alignment changes, essential for detecting extended contours. Consistent with this possibility, previous research has shown that such network modifications lead to better shape processing without necessitating changes in the training visual diet composed of natural image statistics [35,83]. Moreover, [89] demonstrated that an architectural bottleneck in early stages, which simulates information compression leaving the eye via the optic nerve, can naturally develop these early characteristic receptive fields. Finally, there are lateral and recurrent connectivity motifs, such as association fields [48], that could serve to amplify relatively straight contours

causing the feed-forward integrative mechanisms themselves to be biased towards straight or gradually curved contours. Such structural changes in neural networks might be required for DNN models to show alignment with human visual processing that emerges without fine-tuning.

## Conclusion

While our findings do not in any way discount the important role of early lateral or recurrent mechanisms in forming grouped representations in human vision, they do suggest that a feed-forward visual hierarchy with progressively increasing receptive field sizes is capable of playing a significant role in performing fine-grained contour integration and grouping operations. We propose that there is a synergistic interplay between the early local/recurrent mechanisms focused on in prior work, and the feedforward integrative mechanisms discovered in the current work. A key insight is that, despite their large receptive fields, late-stage mechanisms can perform fine-grained integration operations underlying contour-integration, in addition to the longer-range grouping operations underlying uncrowding. Interactions between these early and late mechanisms may result in more sophisticated mechanisms of perceptual grouping than previously explored, or alternatively it maybe be the case that lateral/recurrent operations serve primarily to "distill" the inputs to feed-forward integrative mechanisms. Deep neural network models offer a powerful testbed for instantiating and examining such complex processes, and using these tools to focus on the interplay between levels of processing promises deeper insights into the formation of holistic perceptual representations in both biological and artificial systems.

## Methods

### Ethics statement

This study was approved by the IRB at Harvard University under IRB20–0960 and was supported by NSF PAC COMP-COG 1946308 awarded to G.A.A. All participants completed the study online, and clicked a button to consent to participating in the research study.

### Stimuli generation

**Gabor elements.** We used gabor elements as the fundamental local elements in the contour displays. These elements are sinusoidal gratings modulated by a gaussian envelope. The sinusoidal wave component represents the grating pattern, created by the interaction of the spatial wavelength, orientation, and phase offset, while the gaussian envelope, a bell-shaped curve that modulates the visibility of the sinusoidal wave making it fade out gradually from the center to the edges, is parametrized by its standard deviation. The resulting Gabor element combines these two components, encapsulating spatial frequency and orientation information within a confined spatial region, making them ideal for investigating visual mechanisms due to their compatibility with the receptive fields of neurons in the visual cortex. These elements are characterized by the following equation:

A spatial wavelength ($\lambda$) of 8 pixels was chosen. This value dictates the wavelength of the sinusoidal wave – i.e., the number of stripes. A smaller $\lambda$ value results in more stripes, whereas a larger $\lambda$ value results in fewer stripes. The sinusoidal wave has a phase offset of -90 degrees. This parameter affects the positional shift between the dark and light stripes in the wave, hence controlling the symmetry of the gabor pattern. The standard deviation, which controls the spread of the gaussian envelope and determines how quickly the sinusoidal wave will fade out in the gabor pattern, was set to 4 pixels. Each gabor element is embedded in an image patch of size 28 x 28 pixels. The orientation ($\theta$) of the Gabor elements is variably controlled and generated algorithmically (see the **Contour Path Algorithm**). The orientation decides the angle at which the stripes in the Gabor pattern are tilted. The orientation is particularly crucial as it determines the angle of the sinusoidal wave within the Gabor patch. The contrast was maintained so as to maximally ensure optimal visibility and effectuality in probing the visual system.

**Contour path algorithm.** The contour path generation follows the algorithm initially proposed by [48]. The image, sized at 512x512 pixels, is divided into a 16x16 grid, resulting in cells of 32x32 pixels each. Hence the image grid contains 256 total cells, and each cell contains a gabor element. See S1A Fig, bottom.

The path algorithm sets up 13 anchor points, where 12 gabors will be placed between the anchor points along the path defined by the anchor points. A starting anchor point is randomly determined, situated 64 pixels from the image's center. A vector is projected from the starting point towards the image's center, adjusted by an angle of ± β. The subsequent anchor point is located at a distance, D, from the starting point, where D is set to 32 pixels (one grid). A Gabor element is designated to be placed between these two points, marking the specific grid as occupied, and determining the angle of orientation along the path. If this grid is already pre-occupied, the vector's length is incremented by ΔD, which is set to 25% of D (8 pixels). Using the previously identified anchor point as a reference, the vector is once again adjusted by ± β. An offset of Δβ, which is a random integer between -10 and 10 degrees sampled from a uniform distribution, is added to these adjustments. This process is repeated to determine the next anchor points and Gabor element positions. This procedure is carried out until the 13th anchor point is identified. The β value is varied parametrically in our experiments: a lower β yields straighter contours, while a higher β produces more curved paths. S1A Fig illustrates the parameters involved in this path generation process. The figure below it depicts the placement of anchor points and the targeted locations for gabor elements.

**Contour stimuli.** In our study, we generated matched pairs of images: ones with an extended contour ('contour-present' image) and others without the extended contour ('contour-absent' image). For a 'contour-present' image, 12 Gabor elements are positioned based on the locations determined by the path generation algorithm, with their orientations set to the orientation of the line connecting the two surrounding anchor points. Conversely, in the matched 'contour-absent' image, gabor patches are placed in identical locations, but their orientation is randomized. This procedure is depicted in S1B Fig. Finally, we place background Gabor elements in the unoccupied cells, with random orientations. For a matched pair of stimuli, the background elements were identical, so the displays can only be distinguished based on the orientation-alignment of the contour elements. S1C Fig displays a representative pair of contour-present and contour-absent images.

When spatially positioning each Gabor into a grid cell, path elements were not centered in the grid, but were instead placed at the midpoint of the two anchor points, leading to spatial jitter within each cell. For background elements, spatial jitter was included, displacing elements by up to 9 pixels in both the x and y directions. Note that the spatial position and orientation of each background gabor element was identical between the matched 'contour-present' and 'contour-absent' images.

**Contour and background masks.** Binary masks are made by thresholding the gaussians at the specific location where a gabor element was places. This is done separately for the contour and background elements giving 2 masks. The gaussians vary in a range of 0–255. A threshold of 140 was chosen to create the masks.

## Contour-detection linear probing

**Contour readout.** We examined the Alexnet's contour integration capability by training a separate linear classifier on outputs from each of its 21 layers, as depicted in Fig 2A. Each layer's output was probed to quantify contour-detection capacity. To train the linear classifier, we first flatten the outputs from a layer and pass it as input to the readout classifier. This classifier linearly transforms the input and outputs if a contour is present or not, represented as two values in a one-hot encoding format. The readout classifier has the weight parameter which is of shape [size of output from a layer, 2] and bias terms for each of the two output nodes. For the random and pretrained models (gray and blue points in Fig 2A), we kept the backbone weights frozen—randomized for the former and pretrained (for object recognition on Imagenet) for the latter—during the contour readout training phase. Conversely, finetuned models (blue points in Fig 2A) were trained end-to-end, starting with pretrained weights at initialization, thus providing an assessment of the network's architectural capacity to support contour detection.

**Training stimuli.** We constructed a comprehensive dataset (for training purposes) comprising pairs of contour images. Each pair was created for 91 distinct beta values (global curvature) ranging from 0 to 90 degrees, generating 2,500 pairs per beta value. With the alpha value (local orientation) fixed at 0 degrees, this dataset encompassed a total of 455,000 images. The parameters for the gabors and contour path adhered to the specifics mentioned previously. Specific images were chosen to tailor the 'visual diet' for our contour detection models —those designed to process a broad range of curvatures (shown in Fig 2) were trained on 4,992 images spanning six beta values (0, 15, 30, 45, 60, and 75 degrees) to ensure uniform distribution. Each beta condition was represented by 832 images, yielding 2,496 pairs of contour stimuli (contour-present and contour-absent images). In contrast, the narrowly-focused models (depicted in Fig 5) were trained on approximately 5,000 images from a single or a limited range of beta values.

**Test stimuli.** Our held-out test dataset mirrored the diversity of the training set, featuring 50 pairs per beta value, for all the 91 beta values, totaling 9,100 images. Models trained on a broad spectrum of beta values were evaluated against 600 images distributed equally across these values (0, 15, 30, 45, 60, and 75 degrees). Narrowly-focused models were tested on 50 pairs strictly from their training beta values, aligning the testing to the training approach and ensuring the evaluation's fidelity to the model's training experience.

**Hyperparameters for training contour readouts.** We used stochastic gradient descent (SGD) as the optimizer with a learning rate of 0.0001, momentum of 0.9, and no weight decay to train the readout models. The training loss was computed using a cross-entropy loss function, which is appropriate for this binary classification task. The batch size was set to 8 samples in each batch. We used Pytorch's OnecycleLR as the learning rate scheduler which allowed the learning rate to vary over the course of training. The maximum learning rate was set to 0.0001, with 40% of the total training steps dedicated to increasing the learning rate, followed by a gradual decrease. The scheduler and optimization to train readout models was configured for 100 total epochs, with the learning rate adjusted dynamically based on the training progress.

**Confidence interval for test stimuli.** To evaluate model performance and variability, we calculated 95% confidence intervals (CIs) for contour detection accuracies. These intervals were computed using predictions from a held-out test set of N = 600 images, as described in Section C. First, the model classified each test image as either containing a contour or not. Each prediction was then evaluated for correctness, generating a binary list (1 for correct, 0 for incorrect). The mean accuracy and the standard deviation were calculated across all 600 test images using this correctness list. Finally, the 95% CI was computed using the standard method based on the normal approximation (mean ± 1.96 x standard error of the mean).

## Model sensitivity to alignment of local elements

**Saliency maps via guided backpropagation.** To compute the saliency maps, we pass an image forward through the model and get the output predictions. This forward pass comprises passing the image though the deep net backbone up to a certain layer and then passing the activations from that layer through the attached readout layer. We then compute the gradient, with respect to the output prediction being "contour present", and backpropagate it through the network. A critical manipulation, however, is to modify this backpropagated gradient, at every Relu layer that is involved in the forward pass, by zeroing out all the negative gradients. This ensures that only positive contributions will finally be interpreted. We apply this modification repeatedly at every intermediate relu layer till the gradient is backpropagated at the pixel-level input. At this stage, the gradients can be visualized as saliency maps, providing us with insights about which parts of the image finally influences the model to predict the presence of a contour.

**Sensitivity to location and alignment of local contour elements.** For every image, we use the binary masks and the saliency map to separate the saliency values into two distributions, one group for the contour elements and another group for the background elements. The difference in the mean of these two distributions can be interpreted as the sensitivity of the model to the contour gabor patches. This value is bounded between -1.0 and 1.0 where a value of -1.0 means completely attending to the background patch relative to the contour and a value of 1.0 would mean completely attending to the contour relative to the background.

While the difference in means provides a straightforward understanding of model sensitivity towards foreground versus background elements, it lacks depth in capturing the nuances of their respective saliency distributions. More specifically, solely relying on mean differences can be misleading, as it doesn't account for the ranking and variance within these distributions. A mean-centric measure could yield similar values for distributions that are inherently different, potentially obscuring significant variations in sensitivity that are crucial for our analysis. Hence, we used another metric, called A' which is derived from the area under the Receiver Operating Characteristic (ROC) curve, providing a scalar value between 0 and 1 that indicates the probability that a randomly selected contour element's saliency value is ranked higher than a randomly selected background saliency value. In other words, a value of 0 suggests that the distribution of contour saliency values is consistently ranked lower than that of the background values, indicating no sensitivity to the contour elements. Conversely, a value of 1 suggests perfect sensitivity, with contour elements consistently ranked higher than background elements. This measure effectively characterizes the separability and overlap between the contour and background elements' saliency distributions, providing a more granular understanding of the model's attention mechanism without making assumptions about the normality of the underlying distributions.

To measure the model's sensitivity to local alignment of contour elements, we measure the contour location sensitivity separately for image pairs which contain aligned or misaligned contour elements but at the exact same location. The difference of each pair lets us asses how much a model is influenced by the alignment of contour elements when detecting an extended contour. We measure the average difference (along with and the 95% CI of the differences to quantify the sensitivity to alignment of the contour elements.

## PinholeNet models

These networks, akin in architecture to Alexnet, deviate primarily by the removal of maxpool operations within their convolutional layers. Notwithstanding this alteration, the count of channels post each convolutional layer (C1 through C5) remains constant across all iterations of PinholeNets, maintaining at 64, 192, 384, 256, and 256 channels respectively. Each convolutional layer engages in a convolutional operation followed by a relu non-linearity. Where these models diverge is in their kernel size and stride during convolutional operations, rendering their deep net units with a restricted field-of-view, culminating in effective receptive field sizes (in the C5 units) of 11, 17, 31, and 33 pixels for the P11, P17, P31, and P33 models respectively (refer to Fig 3A for clarity). These models are structurally similar to Alexnet but differ primarily in the omission of maxpool operations in their convolutional layers. Despite this change, the number of channels after each convolutional block (from C1 to C5) remains consistent across all PinholeNet versions, with 64, 192, 384, 256, and 256 channels in the C1, C2, C3, C4, and C5 blocks respectively. Each block engages in a convolutional operation followed by a relu non-linearity. The distinct characteristic of these models is the variation in kernel size and stride during convolutions, rendering their deep net units with a restricted field-of-view. This results in the receptive field sizes of 11, 17, 31, and 33 pixels for deep net units in the fifth convolutional block. (see S4A Fig for a visual depiction of the architecture).

## Behavioral experiment

**Stimulus design and beta conditions.** The human experiments utilized five beta values (15°, 30°, 45°, 60°, and 75°) following the psychophysical experiments conducted by [48], where each beta value represents the global curvature of the contour path. Note that the displays used in the behavioral experiment were distinct from the held-out test set used for computing model validation performance (which were generated from the same set of beta parameters, and also included Beta 0° condition, unlike the human psychophysics experiment).

**Experimental design.** Each participant undertook a 2-IFC (Two-Interval Forced Choice) contour detection task, as illustrated in Fig 4B. Participants were prompted with a 'Get Ready!' indicator, followed by a blank screen that lasted for 500 ms. Sequentially, two displays were presented, each for 500 ms, interspersed by a blank screen with a duration of 1000 ms. Their task was to discern which of the two displays (Display 1 or Display 2) contained an extended contour by

registering their response via keypress. The first display was randomly rotated (0°, 90°, 180°, or 270°) and the second display was further rotated by 90°, 180°, or 270° from the first. This global rotation ensured that background elements were not systematically aligned between displays, thereby reducing the possibility of participants noticing that the displays were matched and relying on change-detection strategies. Over the course of the experiment, each participant encountered 200 total trials from one of the 5 file splits (i.e., each trial containing 200 paired contour displays). These trials are counterbalanced across the 5 beta conditions and the 2 displays (i.e., equal number of contour-present images are shown in Display 1 and Display 2). The experiment is segmented into five blocks, with each block presenting 40 counterbalanced trials (spanning all beta values and display conditions, hence 4 trials spanning the 10 total conditions). A resting period followed each block, with participants advancing to subsequent blocks by pressing the spacebar when ready. Each block is further subdivided into two mini-blocks. Both mini-blocks contain a counterbalanced set of trials, and they transition without any intervening breaks. The rationale behind this design is to ensure that while conditions are perfectly counterbalanced throughout the experiment and within each block, their presentation within a block isn't skewed. Specifically, without the use of these mini-blocks, there could be unintended randomness where certain conditions predominantly appear at the beginning or at the end of a trial block.

**Exclusion criteria.** Initially, participants were recursively removed until the remaining participants' overall accuracy (or mean percent correct) for all the images that the participants viewed lay within 2.5 times the standard deviation. This step led to the exclusion of 11 participants. S5A Fig, top shows the mean overall accuracy before and after the exclusion. However, simply aligning with overall accuracy might not guarantee that individual participants reflect the general trend observed across all beta conditions. To address this, for each participant, we calculated their mean accuracy across all beta conditions, resulting in a five-dimensional vector (each representing one of the 5 beta conditions). We then assessed the correlation between an individual's vector and the average of the vectors across the remaining participants (excluding the target participant) to measure the degree to which a particular participants' performance trended with the group. Any participant whose correlation values lay outside 2.5 times the standard deviation of these correlation values was recursively excluded, leading to the further exclusion of 19 participants. S5A Fig, bottom shows the mean overall accuracy before and after the exclusion This analysis resulted in the exclusion of a total 30 participants from the initial set of 108 participants, giving a set of 78 participants who met both our overall accuracy and beta-condition trend criteria. The participants were recruited using Prolific (www.prolific.com). A post-hoc analysis with all the participants (n = 108) shows all the same patterns and very minimal numeric differences (S14 Fig).

## Contour signal strength in humans and models

To quantify the contour signal in humans and models, we employed a signal detection theory (SDT) framework which allowed us to discern the strength and reliability of contour signals across a range of image displays.

**Human percent correct.** For human participants, the signal detection framework was operationalized by calculating the percent correct across trials. Each participant was subjected to a 2-interval forced-choice (2-IFC) task, wherein they were presented with pairs of displays (one contour-present and one contour-absent) and asked to identify the display containing the extended contour. Their performance was recorded as a binary correct/incorrect outcome for each trial. The human signal strength, therefore, was represented by the percentage of correct responses, with higher percentages reflecting greater sensitivity to contour presence. These individual performance metrics were aggregated to form a group-level performance profile against which the model's signal strength could be compared.

**Model contour signal.** The model's signal strength was determined by assessing the activation levels at the "contour present" node within the model's architecture. We measured the activation responses of the "contour present" node when processing image pairs from the psychophysics dataset. For each pair — one image with a contour (contour-present) and one without (contour-absent) — we calculated the difference in activation levels at the "contour present" node. This node's response provides an index of the model's confidence in the presence of a contour within the image. The signal

strength for each image pair was evaluated by plotting these differential activations on a 2D scatter plot, where the x-axis represented activations for contour-present images, and the y-axis for contour-absent images. The critical measure of signal strength is the distance of each point from the identity line (diagonal) on this plot. Points farther from the diagonal represent a greater discrepancy between the model's responses to contour-present versus contour-absent images, thus indicating a stronger contour signal. This distance-from-diagonal metric effectively serves as a signal-to-noise ratio, capturing the model's ability to discriminate between images with and without contours (see S7 Fig).

## Assessing other network architectures

**Pre-trained models.** We used features from two pre-trained architectures: ResNet-50 and a Vision Transformer. The ResNet-50 model was adapted from PyTorch's torchvision library [90] and was pre-trained on ImageNet for object categorization. The Vision Transformer [67] was adapted from the timm library [69] and was also pre-trained on ImageNet for object categorization, but with additional augmentation and regularization techniques [68].

**Model input and preprocessing.** For the ResNet-50 model, we retained the original size of the contour displays (512 x 512). The adaptive average pooling layer in ResNet-50 allows for flexibility in input resolution, enabling the use of these larger images even though ResNet-50 was originally trained on 224 x 224 images. The images were normalized using the mean ([0.485, 0.456, 0.406]) and standard deviation ([0.229, 0.224, 0.225]) of the ImageNet dataset for each RGB channel.

For the Vision Transformer, we had to resize the images to meet the architectural constraints of the model, which require an input of 224 x 224, divided into 196 patches (14 x 14 grid), with each patch being 16 x 16 pixels. To achieve this, we first resized the 512 x 512 contour images to 224 x 224 pixels using bicubic interpolation, which ensures smooth scaling by considering the nearest 16 pixels during resizing. This method preserves the details of the Gabor elements while maintaining smooth transitions in pixel intensity. The images were normalized using a mean of [0.5, 0.5, 0.5] and a standard deviation of [0.5, 0.5, 0.5] for each channel (RGB). This was done to maintain consistency with the original pre-processing used during the model's pre-training. This resizing process reduced the Gabor elements from 28 x 28 pixels (originally embedded within a 16 x 16 grid in the 512 x 512 pixel image) to approximately 12 x 12 pixels (in the 224 x 224 pixel image broken down into 14 x 14 patches), resulting them to approximately fall within individual patches processed by the Vision Transformer.

**Training linear readout for contour detection.** For both ResNet-50 and the Vision Transformer, we froze the pre-trained backbone features and trained linear probes to determine whether contour presence could be detected from intermediate layers. For ResNet-50, the output of each ReLU block and the final pooling layer was used to train the linear probes, while for the Vision Transformer, the linear probes were trained using the class token representation (containing 768 features) from each of the 12 transformer blocks.

The hyperparameters and contour "diet" used for training the linear readouts were identical to those used for the AlexNet model layers. The task involved identifying extended contours formed by Gabor elements embedded within a grid of randomly oriented Gabor elements. For each model, accuracy in detecting contour presence was measured by training linear readouts at each layer or block. By keeping the backbone features fixed and only updating the weights in the linear readout layer, we were able to assess the extent to which contour detection was supported by the learned representations in the intermediate layers of these pre-trained models.

## Assessing the feedforward architecture on visual uncrowding

To test the capacity of a purely feedforward architecture to exhibit uncrowding, we implemented a vernier acuity task modeled after [42,45].

**Stimuli.** The stimuli were generated using a procedure adapted from the publicly available code from [42]. All stimuli were generated on a 227x227 pixel canvas. The vernier target consisted of two vertical lines, each 20 pixels long and 2

pixels wide, separated by a vertical gap of 2 pixels. Flankers were one of five shapes (square, circle, hexagon, octagon, or diamond), each with a size of 19x19 pixels. All images had a noise level of 0.1. For training, batches were generated as completely "non-overlapping" stimuli, where each image contained both a vernier and a flanker configuration placed in spatially separate locations. For testing, batches were generated as completely"overlapping" or "crowded" stimuli, where the vernier target was centered within a configuration of 1, 3, 5, or 7 flankers.

**Training the feedforward architecture.** We used the VGG19-BN architecture, pretrained on ImageNet, as our feedforward architecture. A linear readout containing two output nodes to perform the binary (left/right) vernier offset discrimination task was attached to the avgpool layer after the convolutional backbone. The backbone was initialized with ImageNet weights and either kept frozen or fine-tuned end-to-end for training on the vernier offset discrimination task. For both conditions, models were trained for 200 epochs using the AdamW optimizer, a batch size of 32, and a Cross-Entropy Loss function. We used a OneCycleLR learning rate scheduler with a maximum learning rate of 0.0001. The models were trained on 640k stimuli. We tested model performance on 64k images in each vernier-flanker configuration.

## Supporting information

**S1 Fig. Contour stimuli generation. (A)** Illustration of the contour path algorithm, showing the contour elements (Gabors). These gabors are 28 pixels in size and are about 32 pixels apart with an angular difference of ± (random) β degrees between consecutive contour elements. The bottom panel shows a 512 x 512 pixels grid highlighting the locations of these elements with the first element of the contour always being at a 64 pixel distance from the center. **(B)** Example of placing contour and background elements for contour-present (left) and contour-absent (right) conditions. The top row shows the placement of contour elements and the bottom row shows the placement of background elements. **(C)** Example contour-present and contour-absent stimuli.
(TIF)

**S2 Fig. Sensitivity analysis of contour elements in fine-tuned models. (A)** Saliency maps illustrating the relevance of each pixel in detecting a contour within an example image. The top row shows the contour-present image, the corresponding contour mask, and the background mask. The bottom row displays saliency maps for a fine-tuned model with readouts from the Conv2 layer and the Avgpool layer, respectively. **(B)** Location sensitivity of contour elements across different readout layers. The upper plot shows the difference of means, while the lower plot shows aprime (accuracy time) for all readout models. Error bars denote 95% confidence intervals. **(C)** Alignment sensitivity of contour elements, showing the difference between misaligned and aligned pairs across different readout layers. The inset highlights the sensitivity to contour elements (computed using the aprime measure) for locally misaligned and locally aligned elements, with a focus on the difference in sensitivity (indicated by the red arrow and circled region). Error bars represent 95% confidence intervals.
(TIF)

**S3 Fig. Model performance is sensitive to disruptions in alignment beyond a simple heuristic of orientation similarity and proximity. (A)** Schematic showing manipulation of alignment ('good continuation') while maintaining orientation similarity and local proximity between contour elements. This is done by displacing each contour element with an alignment jitter in a direction perpendicular to the local contour path. **(B)** Example stimuli showing a contour with 0 pixels of jitter (left) and 8 pixels of jitter (right). With alignment hitter of 8 pixels, it is much harder to detect the contour in the display. **(C)** A visualization of the isolated contour elements for each alignment jitter value (0, 2, 4, 6, and 8 pixels). **(D)** Performance of broadly-tuned model on the jittered stimuli. **(E)** Performance of narrowly-tuned model (fine-tuned on 20° contours) on the jittered stimuli. Performance systematically declines for both fine-tuned models as alignment jitter increases.
(TIF)

**S4 Fig. Architecture and performance of pinholenet architectures. (A)** Schematic representation of PinholeNet architectures with varying receptive field sizes (P11, P17, P31, P33), highlighting the constrained receptive fields in the convolutional layers. For comparison, a standard AlexNet model is also shown in bottom. **(B)** Top-1 object recognition accuracy on the ImageNet validation set during training for different PinholeNet models (P11, P17, P31, P33) and the standard AlexNet model. **(C)** Contour readout accuracy from the Conv5 layer (C5) for models fine-tuned on contour detection. The plot shows the performance of different PinholeNet models compared to the standard AlexNet model across finetuning epochs. **(D)** Receptive field sizes of intermediate units across different layers (C1 to the 2nd FC layer) in PinholeNet and standard AlexNet models. **(E)** Contour readout accuracy from other layers – the avgpool layer, 1st fully connected layer, and 2nd fully connected layer, for different PinholeNet models (P11, P17, P31, P33) and the standard AlexNet model. The legend denotes the color and line style for each model.
(TIF)

**S5 Fig. Human participant performance in Behavioral experiment. (A)** Distribution of human performance before and after applying exclusion criteria. The left column shows the performance distribution for Criteria 1 and Criteria 2 before exclusion (red histograms), and the right column shows the performance distribution after applying the exclusion criteria (green histograms). **(B)** Subject split across trial files, indicating the number of participants included in different trial files after applying the exclusion criteria. **(C)** Weibull function fits for each participant, showing individual performance curves. Each subplot represents a different participant, with the x-axis indicating the contour curvature condition and the y-axis representing the performance. The blue line represents the Weibull fit, and the red dashed line indicates chance performance.
(TIF)

**S6 Fig. Human performance on contour detection across curvature conditions. (A)** Violin plots showing the distribution of percent correct responses for different global curvature (β) conditions (15°, 30°, 45°, 60°, 75°). **(B)** Percent correct for individual trials sorted based on global curvature and performance. Each color represents a different global curvature condition. **(C)** Example images showing the contour-present stimulus in the easiest trial for each β condition. **(D)** Example images showing the contour-present stimulus in the hardest trial for each β condition.
(TIF)

**S7 Fig. Analysis of contour signal strength and human-model correspondence. (A)** Scatter plot showing activity of the contour-present image on the x-axis and the activity of the corresponding contour-absent image on the y-axis. Activity is computed on the contour-present node of the model. Each point represents an image pair of a trial and the perpendicular distance from the diagonal represents contour signal strength for that trial. **(B)** The top panel shows the model signal strength for each trial, and the bottom panel shows the percent correct responses across humans on the same trials.
(TIF)

**S8 Fig. Human-model correlation across different models reading out from different layers of alexnet and finetuned on a broad range of contour curvatures.** The y-axis represents the correlation between human and model contour signal at the level of individual trials, with the noise ceiling indicated by the gray shaded area at $r = 0.785$. The red dashed line represents zero correlation. The x-axis represents the layer from which readouts were taken with a bottom schematic showing the architecture of the contour readout models.
(TIF)

**S9 Fig. Human-model comparison for different readout models finetuned on specific curvatures (0°, 15°, 30°, 45°, 60°, and 75°). (A)** Human-model correlation of contour signal at the level of individual trials. The y-axis represents the correlation between human and model contour signal at the level of individual trials, with the noise ceiling indicated by the gray shaded area at $r = 0.785$. **(B)** Comparison of humans and model behavior for different global curvature levels.

Each subplot shows the performance of the readout model (blue line) and human participants (black line) across varying β conditions.
(TIF)

**S10 Fig. Human-model correlation across models reading out from different layers of alexnet and finetuned on specific contour curvatures.** Each subplot represents models reading out from a different layer, including Dropout before FC6, FC6, ReLU6, Dropout before FC7, FC7, ReLU7, and FC8. The y-axis represents the correlation between human and model contour signal at the level of individual trials, with the noise ceiling indicated by the gray shaded area at $r = 0.785$. The x-axis represents the training curvature. The schematic above each subplot indicates the architecture of the contour readout models.
(TIF)

**S11 Fig. Contour detection accuracy across different curvatures for narrowly-tuned model.**
(TIF)

**S12 Fig. Contour detection accuracy across different layers of finetuned alexnet, pretrained AlexNet, pretrained ResNet-50, and pretrained vision transformer models.** Accuracy values represent the performance of linear readouts trained to detect the presence of contours from the features of each layer. Fine-tuned Alexnet (left, blue) shows significantly improved accuracy across layers, reaching near-perfect performance in the later layers. Frozen AlexNet (pretrained on ImageNet, second from left) shows moderate contour detection performance, peaking in the deeper layers but not matching the fine-tuned variant. Frozen ResNet-50 (pretrained on ImageNet, second from right) and Frozen Vision Transformer (pretrained on ImageNet, right) exhibits similar accuracy trends to the frozen AlexNet model indicating no enhanced contour detection capacity emergent in pretrained networks. Error bars denote 95% confidence intervals for readout accuracy.
(TIF)

**S13 Fig. Run-to-run variability in fine-tuned VGG19 models on the uncrowding task.** Each panel shows the the vernier discrimination accuracy for a specific training run on the overlapping (crowded) test set, after the model is fine-tuned on non-overlapping train set. **(A)** Runs showing a clear emergent uncrowding signature: a drop in accuracy with one flanker, followed by a systematic improvement as more flankers are added **(B)** Runs that fail to generalize to the crowded test set, with performance remaining at or near chance levels. Note that these models still achieved high vernier discrimination accuracy on the training (non-overlapping) set, their failure is one of generalization, not learning.
(TIF)

**S14 Fig. No impact of participant exclusion criteria on key results.** Correlations between human performance and the models fine-tuned from the avgpool layer, one model for each of the different training curvatures. The left panel represents correlations using the full participant set (n = 108), while the right panel shows correlations for the selected subset (n = 78). The human noise ceiling is at 0.774 (±0.011) for all subjects vs. 0.785 (±0.009) for selected subjects. The controlled models that are narrowly-tuned, demonstrate higher alignment with human behavior, peaking when trained with 20° contours approaching the noise ceiling (all subjects r = 0.766; vs selected subjects: 0.768). In addition, the broadly-tuned model (average pool layer) still shows a lower correlation with human data (all subjects r = 0.186; vs selected subjects: 0.197).
(TIF)

## Acknowledgments

We would like to thank lab members of the Vision Science Lab for their helpful feedback and support during the writing process. Special thanks to Spandan Madan for insightful comments on the manuscript.

## Author contributions

**Conceptualization:** Fenil R. Doshi, Talia Konkle, George A. Alvarez.

**Data curation:** Fenil R. Doshi, George A. Alvarez.

**Formal analysis:** Fenil R. Doshi, Talia Konkle, George A. Alvarez.

**Funding acquisition:** Talia Konkle, George A. Alvarez.

**Investigation:** Fenil R. Doshi, Talia Konkle, George A. Alvarez.

**Methodology:** Fenil R. Doshi, Talia Konkle, George A. Alvarez.

**Project administration:** Fenil R. Doshi, Talia Konkle, George A. Alvarez.

**Resources:** Talia Konkle, George A. Alvarez.

**Software:** Fenil R. Doshi, George A. Alvarez.

**Supervision:** Talia Konkle, George A. Alvarez.

**Validation:** Fenil R. Doshi, Talia Konkle, George A. Alvarez.

**Visualization:** Fenil R. Doshi, Talia Konkle, George A. Alvarez.

**Writing – original draft:** Fenil R. Doshi, Talia Konkle, George A. Alvarez.

**Writing – review & editing:** Fenil R. Doshi, Talia Konkle, George A. Alvarez.

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
