## [Decision Letter · Decision Letter 0]

19 Nov 2024

PCOMPBIOL-D-24-01619A feedforward mechanism for human-like contour integrationPLOS Computational Biology Dear Dr. Doshi, Thank you for submitting your manuscript to PLOS Computational Biology. After careful consideration, we feel that it has merit but does not fully meet PLOS Computational Biology's publication criteria as it currently stands. Therefore, we invite you to submit a revised version of the manuscript that addresses the points raised during the review process. Please submit your revised manuscript within 60 days Jan 19 2025 11:59PM. If you will need more time than this to complete your revisions, please reply to this message or contact the journal office at ploscompbiol@plos.org. Please include the following items when submitting your revised manuscript: * A rebuttal letter that responds to each point raised by the editor and reviewer(s). You should upload this letter as a separate file labeled 'Response to Reviewers'. This file does not need to include responses to formatting updates and technical items listed in the 'Journal Requirements' section below. * A marked-up copy of your manuscript that highlights changes made to the original version. You should upload this as a separate file labeled 'Revised Manuscript with Track Changes'. * An unmarked version of your revised paper without tracked changes. You should upload this as a separate file labeled 'Manuscript'. If you would like to make changes to your financial disclosure, competing interests statement, or data availability statement, please make these updates within the submission form at the time of resubmission. Guidelines for resubmitting your figure files are available below the reviewer comments at the end of this letter. We look forward to receiving your revised manuscript. Kind regards,Roland W. Fleming, PhDAcademic EditorPLOS Computational Biology Hugues BerrySection EditorPLOS Computational Biology Feilim Mac GabhannEditor-in-ChiefPLOS Computational Biology Jason PapinEditor-in-ChiefPLOS Computational Biology **Additional Editor Comments :** The reviewers have raised several quite fundamental issues with the research program and the authors need to think carefully about whether they can satisfactorily address them, or whether it would be better to try submitting to another journal. If the authors are confident that they can, then we welcome the chance for them to address the issues and resubmit.**Journal Requirements:**

At this stage, the following Authors/Authors require contributions: Fenil R. Doshi. Please ensure that the full contributions of each author are acknowledged in the "Add/Edit/Remove Authors" section of our submission form.  

2)We ask that a manuscript source file is provided at Revision. Please upload your manuscript file as a .doc, .docx, .rtf or .tex. If you are providing a .tex file, please upload it under the item type u2018LaTeX Source Fileu2019 and leave your .pdf version as the item type u2018Manuscriptu2019.

4)Please ensure to include the heading "Abstract" in your manuscript. Please ensure all required sections are present and in the correct order. Make sure section heading levels are clearly indicated in the manuscript text, and limit sub-sections to 3 heading levels. An outline of the required sections can be consulted in our submission guidelines here: 

5)Please insert an Ethics Statement at the beginning of your Methods section, under a subheading 'Ethics Statement'. It must include:

i) The full name(s) of the Institutional Review Board(s) or Ethics Committee(s) ii) The approval number(s), or a statement that approval was granted by the named board(s) iii) A statement that formal consent was obtained (must state whether verbal/written) OR the reason consent was not obtained (e.g. anonymity). NOTE: If child participants, the statement must declare that formal consent was obtained from the parent/guardian.]

6)Please upload all main figures as separate Figure files in .tif or .eps format. For more information about how to convert and format your figure files please see our guidelines: 

7) We have noticed that you have uploaded Supporting Information files, but you have not included a list of legends. Please add a full list of legends for your Supporting Information files after the references list.

8) Some material included in your submission may be copyrighted. According to PLOSu2019s copyright policy, authors who use figures or other material (e.g., graphics, clipart, maps) from another author or copyright holder must demonstrate or obtain permission to publish this material under the Creative Commons Attribution 4.0 International (CC BY 4.0) License used by PLOS journals. Please closely review the details of PLOSu2019s copyright requirements here: PLOS Licenses and Copyright. If you need to request permissions from a copyright holder, you may use PLOS's Copyright Content Permission form.

Potential Copyright Issues:

i) Please confirm (a) that you are the photographer of 1A, or (b) provide written permission from the photographer to publish the photo(s) under our CC BY 4.0 license. ii) Figures 1A, 2A, and 6B. Please confirm whether you drew the images / clip-art within the figure panels by hand. If you did not draw the images, please provide (a) a link to the source of the images or icons and their license / terms of use; or (b) written permission from the copyright holder to publish the images or icons under our CC BY 4.0 license. Alternatively, you may replace the images with open source alternatives. See these open source resources you may use to replace images / clip-art: - https://commons.wikimedia.org - https://openclipart.org/

9) Please amend your detailed Financial Disclosure statement. This is published with the article. It must therefore be completed in full sentences and contain the exact wording you wish to be published.

1) State what role the funders took in the study. If the funders had no role in your study, please state: "The funders had no role in study design, data collection and analysis, decision to publish, or preparation of the manuscript.". If you did not receive any funding for this study, please simply state: u201cThe authors received no specific funding for this work.u201d

**Reviewers' comments:**Reviewer's Responses to Questions

**Comments to the Authors:**

Reviewer #1: The authors explore the possibility that path detection and contour integration can be achieved through purely feedforward processes by finetuning an ImageNet trained AlexNet to do the path detection task. Results suggest that paths can be accurately detected by finetuning networks for the task but that networks trained purely for object recognition do not accurately detect these paths. The authors also report evidence that DCNNs’ performance on the path detection task (as a function of turn angle between consecutive Gabors) most closely correlates with humans when networks are finetuned with a turn angle of 20 degrees. Better-performing neural networks like ResNet and ViT do not get path detection for free from their more sophisticated architecture.

I liked this paper quite a bit and think it has great potential to be widely cited by vision researchers. I agree with the authors’ approach of using neural networks not as models of visual perception but as tools for testing what is computable with the kinds of architectures by which networks are built. The authors offer a compelling existence proof that feedforward mechanisms are on their own sufficient for humanlike contour integration. I do have a few suggestions that I think would strengthen the article and that I would like to see the authors address before I recommend it for acceptance.

First, I was curious if the authors played with minimally misaligning Gabors to test whether small adjustments to spatial position in the local elements have the same large effects on path detection for neural networks as they do for humans. The authors mentioned that they minimally changed displays by changing the orientation of elements in the contour path, but it might be easier to detect that two elements are compatible or incompatible in terms of their orientation than in terms of their spatial alignment. If two contours have compatible orientation but their positions are slightly jittered so that they do not form a path, will networks fail to see a path as humans do?

I would also be interested to see a comparison in the effect of Gabor density on human and DCNNs’ sensitivity to the contour path. I know that the authors’ control stimuli equated density, but I am curious about the role density plays in displays with a perceptible path. Does path detection fall off more or less quickly as a function of inter-element distance in neural networks compared with humans?

Minor:

I was a little concerned by the authors’ criteria for excluding data from the human experiment on the basis of its dissimilarity to other participants. Do any of the important findings from the paper change if the authors include these excluded participants?

I am unclear on how the authors computed the 95% confidence intervals in Figure 2. Perhaps I missed this in the Methods. Could they explain this more?

Line 193: Figure 2B should be Figure 2A

Line 225: “The model relies on fine-grained collinearity between contour elements to isolate the contour from the background.” Aren’t there many elements that are not collinear that still organize together into a path? Unless I am misunderstanding, reliance on collinearity seems to strict for what humans and networks are both doing here.

Line 293: There is a typo here. “…while still able to succeed at object recognition” � “but still being able to succeed at object recognition”.

Line 362: “were able to accurately detect relative straight contours” � “were able to accurately detect relatively straight contours”

Line 427: “…that we are not directly fit the model” should be “that we did not directly fit the model” or “that we are not directly fitting the model”

Line 471: “In an auxiliary analyses” should be “In auxiliary analyses” or “In an auxiliary analysis”

Reviewer #2: This study examines the applicability of feedforward network architectures to a well-known psychophysical paradigm (informally known as "snake stimuli") that was popular in the 1990s for studying contour integration in human vision. The main finding is that these architectures can perform the task associated with this paradigm. Furthermore, under some conditions, they can mimic some aspects of human behavior.

When I first read the abstract of this paper, I was excited: it suggested to me that this paper challenged a bold proposal put forward by two articles, one in PLoS Comp Biol (this journal) and one in Vision Research, by Doerig et al:

Doerig A, Schmittwilken L, Sayim B, Manassi M, Herzog MH (2020) Capsule networks as recurrent models of grouping and segmentation. PLoS Comput Biol 16(7): e1008017).

A. Doerig, A. Bornet, O.H. Choung, M.H. Herzog, Crowding reveals fundamental differences in local vs. global processing in humans and machines, Vision Research, Volume 167, 2020, Pages 39-45.

The second article makes a claim that is directly relevant to the counter-claim made in the present manuscript: "Our results provide evidence that ffCNNs cannot produce human-like global shape computations for principled architectural reasons." (from the Abstract of Doerig et al. Vision Research). The "ff" in ffCNNs stands for feedforward.

Doerig et al proposed that ``uncrowding,'' a counterintuitive phenomenon in human perceptual grouping, cannot be explained by feedforward architectures. Because the perceptual grouping processes that support uncrowding also involve contour integration and segmentation, and because the claims by Doerig et al are closely related to those made in the present ms, this prior work seems directly relevant to the submission under review.

Disappointingly, when I accepted to review it on the back of that initial excitement, I realized that this study focuses instead on a different phenomenon in perceptual grouping, and one that has lost much of its interest for a good part of the past 20 years. It was popular before 2000, but only sporadically seen in the perceptual literature these days. There is no mention of prior work on uncrowding, a much more interesting phenomenon that continues to challenge current ideas on grouping/segmentation and their relation to vision models in machine learning.

More to the point with specific reference to the central claim of this manuscript, nobody in 2024 with minimal familiarity with the work that has been carried out over the past two decades would think it reasonable to advance the notion that *no* DNN can solve snake stimuli, or more generally perform good continuation, or more specifically reproduce a human behavioral trace consisting of 5 data points (Figure 5F). When the authors stage this position as the accepted claim they set out to challenge, they are setting up a straw-man argument: they refer to a paper published by Field et al. in 1993 (lines 82-84), i.e. 30 years ago. Nobody today would think it reasonable to make such a claim (and even what Field et al. meant is not exactly what the authors of this paper claim they meant, but that is not the sticking point here).

Technically speaking, the paper is perfectly adequate: it goes through standard alignment procedures for testing compliance with human behavior. There is no material that strikes this reader as particularly innovative, but nothing to complain about either: everything is executed well and in line with current standards. The problem is that, as I hinted at above, the primary finding is completely unsurprising: of course feedforward architectures can do this, of course they can do it more or less well depending on how much good continuation bias is injected into the learning diet, whether directly (fine-tuning) or indirectly (pre-training), of course having increasingly larger RFs is conducive to solving a problem that requires long-range integration over space. What is surprising about these results? Perhaps that, under some interpretative extrapolation of what others may have thought (or not) three decades ago, they now prove them wrong in 2024? I fail to understand the relevance of this result today.

To be absolutely clear: I think these results should be published, but (in my opinion) not in PLoS CB, or at least not in their current form (see further below for suggestions). The study is technically well-conducted, and it explores a psychophysical phenomenon that has not been addressed before in this way, but if we ask precisely why it hasn't been studied in this way by others before, the answer in my mind is clear: in 2024, it is simply not interesting to choose this particular phenomenon to exercise the power/capability of DNNs. That is why previous investigators interested in this question have set out to demonstrate the power and limits of DNNs with reference to much more challenging problems, such as uncrowding (see material cited above).

In conclusion, and in an effort to provide constructive criticism and offer suggestions for improvement, if the authors are able to revise this study and demonstrate that feedforward architectures can capture human behavior for a challenging problem like uncrowding, then I would be happy to support (in principle and following evaluation that the claim is justified) publication in PLoS CB. But as the manuscript stands now, i.e. limited to the particular phenomenon of contour integration in snake stimuli, I don't see how this study can make the threshold for a journal like PLoS CB, at least insofar as my (high) opinion of this journal is concerned.

Reviewer #3: The authors assess the extent to which a standard CNN captures characteristics of human contour-integration, and what modifications are required to coax the net into showing more human-like behavior. The key results are that standard CNNs do not show emergent human-like contour integration and that fine-tuning on the task is required. The authors suggest that their data show that in order to reach high accuracy on contour integration, the typical progression from small to larger receptive fields is important during fine-tuning. Moreover, they suggest that in order to recapitulate the characteristics of human contour-integration in response to different types of contours (i.e., contours with different beta parameters), an inductive bias towards relatively small beta parameters is required during fine tuning.

Overall, I enjoyed reading the manuscript. It’s very well written, mostly clear, and refreshingly cautious in its conclusions. I also believe that the authors’ approach is useful and worth being published. However, I have a few comments/concerns, which the authors will hopefully find useful.

General/major points:

- I have some reservations about the extent to which the authors really show that fine-tuned models show human-like performance (see below). However, for the sake of the argument, let’s assume that their fine-tuned models do show human-like performance. In my view, there is one important point that gets lost throughout the interpretation and discussion: the inductive biases highlighted in this paper don’t lead to emergent human-like performance on contour detection. Rather, these inductive biases have to be used during fine-tuning in order to give rise to human-like performance. I’m a bit at a loss about what that result tells us about human vision, machine vision, or the relationship between the two. The authors show that in order to achieve human-like performance, they have to use certain inductive biases during fine-tuning. But the human visual system is (presumably) never fine-tuned in the same way. So, I’m unsure what we are to make of the inductive biases that are required during a type of training to which the human visual system is never subjected. Maybe the authors could clarify their views on this issue? Isn’t it possible that the inductive biases identified here are only important for the specific type of fine-tuning used in this paper, but not for the kind of learning that happens in the human visual system?

- I think the abstract and parts of the discussion are not ideal reflections of the work. For instance, in my view, a sentence such as ‘We identify two key inductive biases that give rise to human-like contour integration in purely feedforward CNNs’ is misleading. It’s not the inductive biases that give rise to human-like contour integration. It’s the fine-tuning on the task that’s required first and foremost! I think the abstract (but also some parts of the discussion) would provide a better summary of the work if the authors focussed more on the finding that current CNNs do not show emergent human-like behaviour, and that for fine-tuning, certain inductive biases are helpful.

- Regarding PinholeNets: I’m not sure how surprising it is that these networks aren’t able to detect contours. They are set up not to be able to process the relevant information from the input due to the restriction in receptive field size. I hope I’m not being unfair but I’m not sure how interesting this finding really is. Could the authors explain in more detail in what sense the finding is not trivial?

- Related to my previous point, I’m wondering whether the authors can really say that a progressive increase in receptive field size is required. Maybe all that’s required is a receptive field size large enough to cover the spatial extent of the contour. For instance, I’m wondering what would happen to contour detection performance if the receptive field for the PinholeNet would be 195x195. I think the existing data already might hint at a possible result: contour detection performance increases steadily from the PinholeNet with the smallest to the largest receptive field (as seen in Fig. 3C, left panel). My expectation is that, as receptive field size increases further, performance would improve as well. If so, I think that’s a clear indication that what matters is not a progressive increase in receptive field size but a size large enough to allow the model to process the relevant characteristic of the input.

- I’m slightly confused about the beta values of the different stimulus levels used in the human observers and the models. I have tried to gather the information together and believe the authors used 6 stimulus levels to assess models (beta of 0 to 75 deg in steps of 15 deg). For human observers, it seems only 5 stimulus levels were used (15 to 75 deg in steps of 15 deg). Is that correct? First, I think this needs to be made much clearer, it really confused me and I unnecessarily wasted time trying to gather the relevant information. More importantly, I think the choice of stimulus levels for the humans might be problematic. Humans are close to perfect with contours that have a beta value of 0. I would be surprised if the model that was trained on contours with a beta value of 20 deg, would match this human performance. However, in order to make strong claims about the training with 20 deg contours, it would be critical to show that the model matches human performance not only on the chosen stimulus levels but also at 0 deg. This is particularly important because it’s not really surprising that a model trained on contours with beta of 20 deg performs best in response to contours with a beta of 15 deg, if the only other tested contours have beta values that are further off. To illustrate my point, assume I have two stimulus levels, one with a beta of 15 and one with a beta of 75. Would it be surprising that a model trained on contours with a beta of 20 performs better with the 15 deg contours than the 75 deg contours. Of course not! But if training with 20 deg beta values leads to better performance on 0 deg stimuli compared to 15 deg stimuli (as I expected would be the case for humans), that would be very surprising in my view. At the very least, I would like to see model performance in response to 0 deg stimuli in the model trained with 20 deg stimuli (in the plot in Fig. 5F).

- Related to my previous point, it seem that this sentence makes little sense: “For example, the model trained on 20 deg curvature might show poor contour detection accuracy at 0 deg, 10 deg, 15 deg, and 25 deg etc.” We are never shown how the model performs at 0 deg, 10 deg or 25 deg.

- I don’t understand how this sentence is linked to the actual analyses: “…our analyses revealed that fine-tuning with an inductive bias towards gradual curvature was required to see more human-like contour detection decisions.” I think whatever the idea here is requires more unpacking (at the least, you need to refer the reader to the specific analysis that supports this statement).

- I’m not sure I understand what the authors have in mind with these sentences:

o 1. “Thus, these results provide a clear direction for future research — to discover new inductive biases that can be built into DNN models so that human-like contour detection emerges without any fine-tuning to the contour task.” Why would that be useful? Why would we want to build these inductive biases into DNNs? In my view, the emergence of human-like contour detection without fine-tuning is not an objective per se. More importantly, the authors haven’t shown that building these inductive biases into DNNs doesn’t require fine-tuning. They have only shown that fine-tuning with specific inductive biases leads to more human-like behaviour on the task for which the model has been fine-tuned.

o 2. “For example, perhaps more human-like contour detection will naturally emerge in in models trained on even more general visual objective (e.g., self-supervised learning or predictive coding over more 3-dimensional object views.).” Maybe. But I’m not sure how this relates to the current paper. Which part of your data supports this idea?

- Line 578: What is an ‘implicit’ representation? I assume the authors think that what is required is an ‘explicit’ representation? If so, could they tell the reader what they have in mind with these terms?

Specific/minor points:

- Methods: I’m not sure it’s ideal to keep the background elements identical across the two intervals. It adds a change-detection element to the task that is not relevant for contour integration. If change detection contributed to task performance, then I would expect that order of presentation had an effect on performance. Specifically, I would expect that trials, in which the no-contour stimulus was presented on the first interval, would lead to higher performance. If the authors agree with this idea, it might be worth running a quick analysis to see whether this prediction pans out. If it does, I think it’s important to discuss the possibility that performance in human observers in this setup is not exclusively driven by contour detection ability.

- Line 209: ‘…the path as compared TO the background…’

- Line 249: ‘…dubbed “PinholeNets”,…’ quotation marks are missing after PinholeNets.

- I would avoid sentences such as this one in the main text and put it in the figure caption: ‘PinholeNets are illustrated in varying shades of blue (lighter shades correspond to models with smaller receptive 278 fields), while the standard Alexnet model is represented in grey.’

- Same as before with sentences like this one: ‘The error bars denote the 95% CI of the mean accuracy for each beta condition bootstrapped across the 78 participants.’

- Fig. 5A: I think the beta value for the first contour is incorrect. If it were really beta of zero, the contour should be straight as far as I understand (except for the delta beta value that’s added). I think the depicted contour is probably one with a beta of 15 deg?

- 'i.e.' should be followed by a comma

- 2-IFC (Two-Alternative Forced Choice) should be (Two-Interval Forced Choice)

- Fig. 5 C and F: you forgot to indicate what the colors mean; why is model performance shown in red in C and in green in F?!? Why are the colors for B and E different.

- Red and green are not ideal choices due to the relatively high prevalence of red-green color blindness.

- Line 548: full stop instead of comma

- Fig. 2: Mention that contour patches are highlighted for illustration.

- Fig. 2A: Why is pre-trained model performance low for deep layers?

- I don’t find Fig. 6 helpful at all. In my view, it contains little informative value.

- Reference 59 doesn’t have any information beyond author names and title

- Line 706: what are ‘current operations’?

**Have the authors made all data and (if applicable) computational code underlying the findings in their manuscript fully available?**

Reviewer #1: Yes

Reviewer #2: None

Reviewer #3: Yes

PLOS authors have the option to publish the peer review history of their article (what does this mean? ). If published, this will include your full peer review and any attached files.

**Do you want your identity to be public for this peer review?** For information about this choice, including consent withdrawal, please see our Privacy Policy .

Reviewer #1: No

Reviewer #2: No

Reviewer #3: No

**Figure resubmission:** While revising your submission, please upload your figure files to the Preflight Analysis and Conversion Engine (PACE) digital diagnostic tool, https://pacev2.apexcovantage.com/. PACE helps ensure that figures meet PLOS requirements. To use PACE, you must first register as a user. Registration is free. Then, login and navigate to the UPLOAD tab, where you will find detailed instructions on how to use the tool. If you encounter any issues or have any questions when using PACE, please email PLOS at figures@plos.org. Please note that Supporting Information files do not need this step. If there are other versions of figure files still present in your submission file inventory at resubmission, please replace them with the PACE-processed versions.
---

## [Decision Letter · Decision Letter 1]

5 Jun 2025

PCOMPBIOL-D-24-01619R1

A feedforward mechanism for human-like contour integration

PLOS Computational Biology

Dear Dr. Doshi,

Thank you for submitting your manuscript to PLOS Computational Biology. After careful consideration, we feel that it has merit but does not fully meet PLOS Computational Biology's publication criteria as it currently stands. Therefore, we invite you to submit a revised version of the manuscript that addresses the points raised during the review process.

In particular, Reviewer#2 is still not convinced that your manuscript brings novelty in the field. They are in particular not satisfied with your answer to their proposal regarding uncrowding, that you quickly treated by simply adding a 10-line text in the Discussion section. We agree with the reviewer that your answer to their question is weak and understand their frustration. Therefore, in the revised version, please make sure to address the comments of Reviewer#1 and to propose a stronger response to Reviewer#2 regarding the ability of feed-forward networks to capture human behavior on more difficult problems than in the current manuscript.

Please submit your revised manuscript within 60 days Aug 05 2025 11:59PM. If you will need more time than this to complete your revisions, please reply to this message or contact the journal office at ploscompbiol@plos.org. Please include the following items when submitting your revised manuscript:

We look forward to receiving your revised manuscript.

Kind regards,

Hugues Berry

Section Editor

PLOS Computational Biology

Hugues Berry

Section Editor

PLOS Computational Biology

**Reviewers' comments:**

Reviewer's Responses to Questions

**Comments to the Authors:**

Reviewer #1: Uploaded as attachment

Reviewer #2: I have read the response by the Authors and the revised manuscript. I remain disappointed: the authors have done nothing of substance to address my concerns, except including a brief paragraph that essentially says: "There is this far more interesting and recent finding in the perceptual literature (uncrowding) that our model cannot explain, but we don't claim that we can explain it, so we leave it for others to figure out". That's fine, but that's not the standard I expect for PLoS CB (which doesn't mean that this would not be ok in some other journal).

In their reply, the authors imply that I was confused about the fact that their model was never trained on human data. Of course I understood this aspect of their submission, and honestly I don't see how that can be a point of surprise: it is what people have been doing for decades, it is the standard approach of setting up an over-parameterized architecture and finding out which parameterization produces metrics that align with human trends. But that wasn't my point. My point was that the phenomenon at hand is not one which, personally, I would have picked to exercise the power of ffCNNs in 2025. In 2010, maybe. But not in 2025. The authors state: "Specifically, we thought that Alexnets would be architecturally incapable of detecting these contours, much like the claim you highlighted in Doerig’s work that “their results provide evidence that ffCNNs cannot produce human-like global shape computations for principled architectural reasons.”" This is a misinterpretation of what Doerig et al meant (and I am not one of those authors, but I understand their work). The "human-like global shape computations" those authors were referring to are those involving release from crowding under complex object-based grouping rules, which extend far beyond contour completion in terms of conceptual significance for contextual vision. If the authors of the paper under review thought that "Alexnets would be architecturally incapable of detecting these contours", in my mind this statement reflects a limitation of their knowledge about this specific area of research in human and machine vision: it would have never occurred to me to even try that, so little is the interest I ascribe to this notion in 2025. In their rebuttal, the authors seem to position snake stimuli alongside uncrowding configurations, as if in both cases they were probing ffCNNs effectively. But this is not at all the case, as I pointed out in my earlier review: it is a straw-man argument. If snake stimuli were on a par with uncrowding, then the authors of this paper would be proving Doerig et al wrong. But they are not, and provide no evidence for that (see para below). And again, I emphasize that I am neither Doerig nor one of his collaborators. I am just a reader who finds the results of Doerig et al interesting, and those in the current submission uninteresting.

What I find more disappointing is that the authors did not make any attempt to explore uncrowding using variations of their approach, if for no other reason, at least to confirm the results/conclusions obtained/reached by Doerig et al (which is always helpful). Instead, they left it as an exercise for others to complete, except the exercise left for others is the very meat of what remains mysterious about contextual vision in 2025.

I do not expect that we can reach a productive point of discussion here, for the following reason: the material under consideration is perfectly clear to everyone involved, so there is no discussion to be had there. The discussion is not even over whether this material is "a worthy and quality scientific contribution" in the words of the authors: yes there is worth here, not in abundance in my opinion (it is one of dozens of papers we now read every year about network alignment with human behavior), but I am not saying there is none. The discussion is over whether this material makes a significant contribution to our understanding of vision for contemporary readers. As far as this contemporary reader is concerned, this is not an interesting PLoS CB paper---in other words, it has not taught me anything I didn't already know and/or expect based on the work that has already been carried out before (e.g. by Doerig et al). However, if other readers think differently, I will be happy to go with that at this stage: we obviously all have different standards for what is suitable in a journal like PLoS CB, and I am not invested in setting a specific threshold for this journal. My only objective was to encourage the authors to do better than they had done, and challenge their approach with truly interesting problems for vision in 2025, rather than focusing on easy problems. They did not take the challenge, and I think this is disappointing, but I am sure somebody else will go the extra mile in the future.

**Have the authors made all data and (if applicable) computational code underlying the findings in their manuscript fully available?**

Reviewer #1: Yes

Reviewer #2: None

PLOS authors have the option to publish the peer review history of their article (what does this mean? ). If published, this will include your full peer review and any attached files.

**Do you want your identity to be public for this peer review?** For information about this choice, including consent withdrawal, please see our Privacy Policy .

Reviewer #1: No

Reviewer #2: No

**Figure resubmission:**
---

## [Editor Report · Decision Letter 2]

4 Aug 2025

Dear Doshi,

We are pleased to inform you that your manuscript 'A feedforward mechanism for human-like contour integration' has been provisionally accepted for publication in PLOS Computational Biology.

Best regards,

Daniele Marinazzo

Section Editor

PLOS Computational Biology

Hugues Berry

Section Editor

PLOS Computational Biology

---

## [Editor Report · Acceptance letter]

PCOMPBIOL-D-24-01619R2

A feedforward mechanism for human-like contour integration

Dear Dr Doshi,

I am pleased to inform you that your manuscript has been formally accepted for publication in PLOS Computational Biology. Your manuscript is now with our production department and you will be notified of the publication date in due course.

With kind regards,

Anita Estes
